# Fisher Flow Matching for Generative Modeling over Discrete Data

Oscar Davis[1][*]    Samuel Kessler[1]    Mircea Petrache[2]    İsmail İlkan Ceylan[1]

Michael Bronstein[1,3]    Avishek Joey Bose[1]

[1]University of Oxford, [2]Pontificia Universidad Católica de Chile, [3]Aithyra

## Abstract

Generative modeling over discrete data has recently seen numerous success stories, with applications spanning language modeling, biological sequence design, and graph-structured molecular data. The predominant generative modeling paradigm for discrete data is still autoregressive, with more recent alternatives based on diffusion or flow-matching falling short of their impressive performance in continuous data settings, such as image or video generation. In this work, we introduce FISHER-FLOW, a novel flow-matching model for discrete data. FISHER-FLOW takes a manifestly geometric perspective by considering categorical distributions over discrete data as points residing on a statistical manifold equipped with its natural Riemannian metric: the *Fisher-Rao metric*. As a result, we demonstrate discrete data itself can be continuously reparameterised to points on the positive orthant of the $d$-hypersphere $\mathbb{S}^d_+$, which allows us to define flows that map any source distribution to target in a principled manner by transporting mass along (closed-form) geodesics of $\mathbb{S}^d_+$. Furthermore, the learned flows in FISHER-FLOW can be further bootstrapped by leveraging Riemannian optimal transport leading to improved training dynamics. We prove that the gradient flow induced by FISHER-FLOW is optimal in reducing the forward KL divergence. We evaluate FISHER-FLOW on an array of synthetic and diverse real-world benchmarks, including designing DNA Promoter, and DNA Enhancer sequences. Empirically, we find that FISHER-FLOW improves over prior diffusion and flow-matching models on these benchmarks. Our code is available at https://github.com/olsdavis/fisher-flow.

## 1 Introduction

The recent success of generative models operating on continuous data such as images has been a watershed moment for AI exceeding even the wildest expectations just a few years ago [69, 22]. A key driver of this progress has come from substantial innovations in simulation-free generative models, the most popular of which include diffusion [38, 66] and flow matching methods [48, 73], leading to a plethora of advances in image generation [16, 30, 55], video generation [21, 14], audio generation [59], and 3D protein structure generation [77, 19], to name a few.

In contrast, analogous advancements in generative models over discrete data domains, such as language models [1, 72], have been dominated by autoregressive models [79], which attribute a simple factorisation of probabilities over sequences. Modern autoregressive models, while impressive, have several key limitations which include the slow sequential sampling of tokens in a sequence, the assumption of a specified ordering over discrete objects, and the degradation of performance

---

[*]Correspondence to oscar.davis@cs.ox.ac.uk.

38th Conference on Neural Information Processing Systems (NeurIPS 2024).

without important inference techniques such as nucleus sampling [39]. It is expected that further progress will come from the principled equivalents of diffusion and flow-matching approaches for categorical distributions in the discrete data setting.

While appealing, one central barrier in constructing diffusion and flow matching over discrete spaces lies in designing an appropriate forward process that progressively corrupts discrete data. This often involves the sophisticated design of transition kernels [8, 23, 2, 50], which hits an ideal stationary distribution—itself remaining an unclear quantity in the discrete setting. An alternative path to designing discrete transitions is to instead opt for a continuous relaxation of discrete data over a continuous space, which then enables the simple application of flow-matching and diffusion. Consequently, past work has relied on relaxing discrete data to points on the interior of the probability simplex [9, 68].

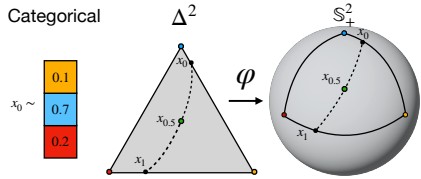

However, since the probability simplex is not Euclidean, it is not possible to utilise Gaussian probability paths—the stationary distribution of an uninformative prior is uniform rather than Gaussian [26]. One possible remedy is to construct conditional probability paths on the simplex using Dirichlet distributions [68], but this can lead to undesirable properties that include a complex parameterisation of the vector field. An even greater limitation is that flows using Dirichlet paths are not general enough to

Figure 1: A geodesic connecting $x_0$ and $x_1$ using the FR metric on $\mathring{\Delta}^2$ and the corresponding path on $\mathbb{S}^2_+$.

accommodate starting from a non-uniform (source) prior distribution—hampering downstream generative modeling applications. These limitations motivate the following research question: *Can we find a continuous reparameterisation of discrete data allowing us to learn a push-forward map between any source and target distribution?*

**Present work**. In this paper, we propose FISHER-FLOW, a new flow matching-based generative model for discrete data. Our key geometric insight is to endow the probability simplex with its natural Riemannian metric—the Fisher-Rao metric—which transforms the space into a Riemannian manifold and induces a different geometry compared to past approaches of Avdeyev et al. [9], Stark et al. [68]. Moreover, using this Riemannian manifold, we exploit a well-known geometric construction: the probability simplex under the Fisher-Rao metric is isometric to the positive orthant of the $d$-dimensional hypersphere $\mathbb{S}^d_+$ [6] (see Figure 1). By operating on $\mathbb{S}^d_+$, we obtain a more flexible and numerically stable parameterisation of learned vector fields as well as the ability to use a familiar metric—namely, the Euclidean metric $\ell_2$ restricted to the sphere, which leads to better training dynamics and improved performance. As a result, FISHER-FLOW becomes an instance of Riemannian Flow Matching (RFM) [26], and our designed flows enjoy explicit and numerically favorable formulas for the trajectory connecting a pair of sampled points between *any* source and target distribution—effectively generalising previous flow models [68].

On a theoretical front, we prove in Proposition 1 that optimising the flow-matching objective with FISHER-FLOW is an optimal choice for matching categorical distributions on the probability simplex. More precisely, we show the direction of the optimal induced gradient flow in the space of probabilities converges to the Fisher-Rao flow in the space of probabilities. In addition, we show in Proposition 2 how to design straighter flows, leading to improved training dynamics, by solving the Riemannian optimal transport problem on $\mathbb{S}^d_+$. Empirically, we investigate FISHER-FLOW on sequence modeling over synthetic categorical densities as well as biological sequence design tasks in DNA promoter and DNA enhancer design. We observe that our approach obtains improved performance to comparable discrete diffusion and flow matching methods of Austin et al. [8], Stark et al. [68].

## 2 Background

The main task of generative modeling is to approximate the target distribution, $p_{\text{data}} \in \mathcal{P}(\mathcal{M})$, over a probability space $(\mathcal{M}, \Sigma, \mathcal{P})$, using a parametric model $p_\theta$. The choice of $\mathcal{M} = \mathbb{R}^d$ appears in the classical setup of generative modeling over continuous domains, *e.g.,* images; while for categorical distributions over discrete data, we identify $\mathcal{M} = \mathcal{P}(\mathcal{A})$, where $\mathcal{A} = \{0, \ldots, d\}$ represents the categories corresponding to an alphabet with $d + 1$ elements. In this paper, we consider problem settings where the modeler has access to $p_{\text{data}}$ as an empirical distribution from which the samples are drawn identically and independently. Such an empirical distribution corresponds to the *training* set

used to train a generative model and is denoted by $\mathcal{D} = \{x_i\}_{i=1}^n$. A standard approach to generative modeling in these settings is to learn parameters $\theta$ of a generative model $p_\theta$ that minimises the forward KL divergence, $\mathbb{D}_{\mathrm{KL}}(p_{\mathrm{data}} || p_\theta)$, or, in other words, maximises the log-likelihood of data under $p_\theta$.

## 2.1 Information geometry

The space of probability distributions $\mathcal{P} = \mathcal{P}(X)$ over a set $X$ can be endowed with a geometric structure. Let $\omega$ be the parameters of a distribution such that the map $\omega \mapsto p_\omega \in \mathcal{P}$ is injective. We note that this map is distinguished from the generative model, $\theta \mapsto p_\theta$, as $\theta$ corresponds to parameters of the *neural network* rather than the parameters of the *output* distribution being approximated. For instance, if we seek to model a multi-variate Gaussian $\mathcal{N}(\mu, \Sigma)$ in $\mathbb{R}^d$, the parameters of the distribution are $\omega = (\mu, \Sigma)$, while $\theta$ can be the parameters of an arbitrary deep neural network.

Our distributions $p_\omega$ are taken to be a family of distributions parameterised by a subset of vectors $\omega = (\omega^1, \ldots, \omega^d) \in \Omega \subseteq \mathbb{R}^d$, with its usual topology. If the distributions $p_\omega$ are absolutely continuous w.r.t. a reference measure $\mu$ over $X$, with densities $p_\omega(x), x \in X, \omega \in \Omega$, then the injective map $\omega \in \Omega \mapsto p_\omega \in L^1(\mu)$ defines a *statistical manifold* (cf. Amari [4], Ay et al. [11]):

$$\mathcal{M}^d := \{p_\omega(\cdot) \,|\, \omega = (\omega^1, \ldots, \omega^d) \in \Omega \subseteq \mathbb{R}^d\}. \tag{1}$$

Note that $\mathcal{M}^d$ is identified as a $d$-dimensional submanifold in the space of absolutely continuous probability distributions $\mathcal{P}(X)$.[2] If $p_\omega(x)$ is differentiable in $\omega$, then $\mathcal{M}^d$ inherits a differentiable structure. We can then define a metric that converts $\mathcal{M}^d$ into a *Riemannian manifold*. Moreover, the parameters $\omega$ are the local coordinates and the map $\omega \mapsto p_\omega$ is a global parameterisation for the manifold.

As for the choice of metric, the minimisation of the forward KL divergence, $\mathbb{D}_{\mathrm{KL}}(p_{\mathrm{data}} || p_\omega)$, under mild conditions, suggests a natural prescription of a Riemannian metric on $\mathcal{M}^d$ [13]. We can arrive at this result by inspecting the log-likelihood of the generative model, $\log p_\omega$, and constructing the Fisher-information matrix whose $(i, j)$-th entry $G(\omega) = [g_{ij}(\omega)]_{ij}$ is defined as

$$g_{ij}(\omega) := \int_\Omega \left( \frac{\partial \log p_\omega}{\partial \omega^i} \right) \left( \frac{\partial \log p_\omega}{\partial \omega^j} \right) p_\omega \, \mathrm{d}\mu, \tag{2}$$

for $1 \leq i, j \leq d$, where $\mu$ is the reference measure on $\Omega$, which must satisfy the property that all $p_\omega$ are absolutely continuous with respect to $\mu$. In this setting, the manifestation of the Fisher-information matrix is not a mere coincidence: it is the second-order Taylor approximation of $\mathbb{D}_{\mathrm{KL}}(p_{p_{\mathrm{data}}} || p_\omega)$ in a local neighborhood of $p_\omega$, in its local coordinates, $\omega$. Furthermore, the Fisher-Information matrix is symmetric and positive-definite, consequently defining a Riemannian metric. It is called the *Fisher-Rao* metric and it equips a family of inner products at the tangent space $\mathcal{T}_{p_\omega}\mathcal{M}^d \times \mathcal{T}_{p_\omega}\mathcal{M}^d \to \mathbb{R}$ that are continuous on the statistical manifold, $\mathcal{M}^d$ (they vary smoothly, in case the map $\omega \in \Omega \mapsto p_\omega \in L^1(\mu)$ is assumed to be smooth). Beyond arising as a natural consequence of KL minimisation in the generative modeling setup, the Fisher-Rao metric is the unique metric invariant to reparameterisation of $\mathcal{M}^d$ (see Ay et al. [11, Thm. 1.2])—a fact we later exploit in §3.2 to build more scalable and more numerically stable generative models.

## 2.2 Flow matching over Riemannian manifolds

A *probability path* on a Riemannian manifold, $\mathcal{M}^d$, is a continuous interpolation between two distributions, $p_0, p_1 \in \mathcal{P}(\mathcal{M}^d)$, indexed by time $t$. Let $p_t$ be a distribution on a probability path that connects $p_0$ to $p_1$ and consider its associated flow, $\psi_t$, and vector field, $u_t$. We can learn a *continuous normalising flow* (CNF) by directly regressing the vector field, $u_t$, with a parametric one, $v_\theta \in \mathcal{T}\mathcal{M}^d$, where $\mathcal{T}\mathcal{M}^d$ is the tangent bundle. In effect, the goal of learning is to match the flow—termed *flow-matching*—of the target vector field and can be formulated into a simulation-free training objective [48, FM], provided $p_t$ satisfies the boundary conditions, $p_0 = p_{\mathrm{data}}$ and $p_1 = p_{\mathrm{prior}}$. As stated, the vanilla flow matching objective is intractable as we generally do not have access to the closed-form of $u_t$ that generates $p_t$. Instead, we can opt to regress $v_\theta$ against a conditional vector field, $u_t(x_t|z)$, generating a conditional probability path $p_t(x_t|z)$, and use it to recover the target unconditional path: $p_t(x_t) = \int_\mathcal{M} p_t(x_t|z)q(z)\mathrm{d}z$. Similarly, the vector field $u_t$ can also be recovered by marginalising conditional vector fields, $u_t(x|z)$. This allows us to state the CFM objective for Riemannian manifolds [26]:

$$\mathcal{L}_{\mathrm{rcfm}}(\theta) = \mathbb{E}_{t, q(z), p_t(x_t|z)} \|v_\theta(t, x_t) - u_t(x_t|z)\|_g^2, \quad t \sim \mathcal{U}(0, 1). \tag{3}$$

---

[2]If, as in our case, we take $X = \mathcal{A} = \{0, \ldots, d\}$, then we can fix $\mu = \frac{1}{d+1} \sum_{i=0}^d \delta_i$, and then $\mathcal{M}^d = \mathcal{P}(X)$.

As FM and CFM objectives have the same gradients [73, 48], at inference, we can generate by sampling from $p_1$, and using $v_\theta$ to propagate the ODE backwards in time. The central question in the Riemannian setting corresponds to then finding $x_t$ and $u_t(x_t|z)$. For simple geometries, one can always exploit the geodesic interpolant to construct $x_t = \exp_{x_0}(t \log_{x_0}(x_1))$ and $u_t(x_t|z) = \dot{x}_t$. Instead of computing the time derivative explicitly, we may also use a general closed-form expression for $u_t$, based on the geometry of the problem: $u_t = \log_{x_t}(x_1)/(1-t)$, cf. [18].

**Notation and convention**. We use $t \in [0, 1]$ to indicate the time index of a process such that $t = 0$ corresponds to $p_{\text{data}}$ and $t = 1$ corresponds to the terminal distribution of a (stochastic) process to be defined later. Typically, this will correspond to an easy-to-sample from source distribution. Henceforth, we use subscripts to denote the time index—*i.e.*, $p_t$—and reserve superscripts to designate indices over coordinates in a (parameter) vector, *e.g.*, $\omega^i \in (\omega^1, \ldots, \omega^d)$.

## 3 Fisher Flow Matching

We now establish a new methodology to perform discrete generative models under a flow-matching paradigm which we term as FISHER-FLOW. Intuitively, our approach begins with the realisation that discrete data modeled as categorical distributions over $d$ categories can be parameterised to live on the $d$-dimensional probability simplex, $\Delta^d$, whose relative interior, $\mathring{\Delta}^d$, can be identified as a Riemannian manifold endowed with the *Fisher-Rao* metric [4, 11, 56]. Additionally, we leverage the *sphere map*, which defines a diffeomorphism between the interior of the probability simplex and the positive orthant of a hypersphere, $\mathbb{S}_+^d$. As a result, generative modeling over discrete data is amenable to continuous parameterisation over spherical manifolds and offers the following key advantages:

**(A1)** **Continuous reparameterisation**. We can now seamlessly define conditional probability paths directly on the Riemannian manifold $\mathbb{S}_+^d$, enabling us to treat discrete generative modeling as continuous, through Riemannian flow matching on the hypersphere.

**(A2)** **Flexibility of source distribution**. In stark contrast with prior work [23, 68], our conditional probability paths can map *any* source distribution to a desired target distribution by leveraging the explicit analytic expression of the geodesics on $\mathbb{S}_+^d$.

**(A3)** **Riemannian optimal transport**. As the sphere map is an isometry of the interior of the probability simplex, we can perform Riemannian OT using the geodesic cost on $\mathbb{S}_+^d$ to construct a coupling between $p_0$ and $p_1$, leading to straighter flows and lower variance training.

In the following subsections, we detail first how to construct the continuous reparameterisation used in FISHER-FLOW §3.1. An algorithmic description of the training procedure of FISHER-FLOW is presented in Algorithm 1. We justify the use of the Fisher-Rao metric in §3.3 by showing that induces a gradient flow that minimises the KL divergence. Finally, we discuss the sphere map in §3.2, and conclude by elevating the constructed flows to minimise the Riemannian OT problem in §3.4.

### 3.1 Reparameterising discrete data on the simplex

We now take our manifold $\mathcal{M}^d = \Delta^d = \{x \in \mathbb{R}^{d+1} | \mathbf{1}^\top x = 1, x \geq 0\}$ as the $d$-dimensional simplex. We seek to model distributions over this space which we denote as $\mathcal{P}(\Delta^d)$. We can represent categorical distributions, $p(x)$, over $K = d+1$ categories in $\Delta^d$ by placing a Dirac $\delta_i$ with weight $p^i$ on each vertex $i \in \{0, \ldots, d\}$.[3] Thus a discrete probability distribution given by a categorical can be converted into a continuous representation over $\Delta^d$ by representing the categories $p^i$ as a mixture of point masses at each vertex of $\Delta^d$. This allows us to write our data distribution $p_{\text{data}}$ over discrete objects as:

$$p_{\text{data}}(x) = \sum_{i=0}^{d} p^i \delta(x - e_i), \tag{4}$$

where $e_i$ are $K = d+1$ one-hot vectors representing the vertices of the probability simplex[4]. While the vertices of $\Delta^d$ are still discrete, the relative interior of the probability simplex, denoted as $\mathring{\Delta}^d := \{x \in \Delta^d : x > 0\}$, is a continuous space, whose geometry can be leveraged to build our

---

[3]We denote, with a slight abuse of notation, the probability of category $i$ by $p^i$, *i.e.*, $\sum_i p^i = 1$.

[4]Note that $e_i \in \Delta^d$ represents Dirac mass $\delta_i \in \mathcal{P}(\mathcal{A})$, thus Eq. 4 means that $p_{\text{data}} = \sum_i p^i \delta_{\delta_i} \in \mathcal{P}(\mathcal{P}(\mathcal{A})) \simeq \mathcal{P}(\Delta^d)$. The traditional form $\sum_i p^i \delta_i \in \mathcal{P}(\mathcal{A})$ is recovered via the identification $\delta_i \mapsto i$.

method, FISHER-FLOW. Consequently, we may move Dirac masses on the vertices of the probability simplex to its interior—and thereby performing continuous reparameterisation —by simply applying any smoothing function $\sigma : \Delta^d \to \mathring{\Delta}^d$, *e.g.,* label smoothing as in supervised learning [70].

**Defining a Riemannian metric**. Relaxing categorical distributions to the relative interior, $\mathring{\Delta}^d$, enables us to consider a more geometric approach to building generative models. Specifically, this construction necessitates that we treat $\mathring{\Delta}^d$ as a *statistical Riemannian manifold* wherein the geometry of the problem corresponds to classical *information geometry* [4, 11, 56]. This leads to a natural choice of Riemannian metric: the Fisher-Rao metric, defined as, on the tangent space at an interior point $p \in \mathring{\Delta}^d$,

$$\forall u, v \in \mathcal{T}_p \mathring{\Delta}^d, \quad g_{\mathrm{FR}}(p)[u, v] := \langle u, v \rangle_p := \left\langle \frac{u}{\sqrt{p}}, \frac{v}{\sqrt{p}} \right\rangle_2 = \sum_{i=0}^{d} \frac{u^i v^i}{p^i}. \tag{5}$$

In the above equation, the inner product normalisation by $\sqrt{p}$ is applied component-wise. After normalising by $\sqrt{p}$ the inner product on the simplex becomes synonymous with the familiar Euclidean inner product $\langle \cdot, \cdot \rangle_2$. However, near the boundary of the simplex, this "tautological" parameterisation of the metric by the components of $p$ is numerically unstable due to division by zero. This motivates a search for a more numerically stable solution which we find through the sphere-map in §3.2. As a Riemannian metric is a choice of inner product that varies smoothly, it can be readily used to define geometric quantities of interest such as distances between points or angles, as well as a metric-induced norm. We refer the interested reader to §B for more details on the geometry of $\mathring{\Delta}^d$.

## 3.2 Flow Matching from $\mathring{\Delta}^d \to \mathbb{S}_+^d$ via the sphere map

The continuous parameterisation of categorical distributions to the interior of the probability simplex, while theoretically appealing, can be prone to numerical challenges. This is primarily because in practice we do not have the explicit probabilities of the input distribution, but instead, one-hot encoded samples which means that we must flow to a vertex. More concretely, this implies that when $t \to 1$, $x_t \to e_i$ for some $i \in [d]$, therefore implying $\|v\|_{x_t} \to \infty$, where $\|\cdot\|_{x_t}$ denotes the norm at point $x_t$. This occurs due to the metric normalisation $\sqrt{p}$, applied component-wise. In addition, the restriction of $v_\theta$ to be at the tangent space imposes architectural constraints on the network. What we instead seek is a flow parameterisation without any architectural restrictions or numerical instability due to the metric norm. We achieve this through the *sphere map*, $\varphi : \mathring{\Delta}^d \to \mathbb{S}_+^d$, which is a diffeomorphism between the interior of the simplex and an open subset of the positive orthant of a $d$-dimensional hypersphere.

$$\begin{aligned} \varphi : \mathring{\Delta}^d &\longrightarrow \mathbb{S}_+^d, \quad p \longmapsto s := \varphi(p) = \sqrt{p}, \\ \varphi^{-1} : \mathbb{S}_+^d &\longrightarrow \mathring{\Delta}^d, \quad s \longmapsto p := \varphi^{-1}(s) = s^2. \end{aligned} \tag{6}$$

In Eq. 6, both the sphere map and its inverse are operations that are applied element-wise. The sphere map reparameterisation identifies the Fisher-Rao geometry of $\mathring{\Delta}^d$ to the geometry of a hypersphere, whose Riemannian metric is induced by the Euclidean inner product of $\mathbb{R}^{d+1}$. It is easy to show that $2\varphi$ (the sphere map scaled by 2) preserves the Riemannian metric of $\mathring{\Delta}^d$, *i.e.,* it is an isometry, and that therefore all geometric notions such as distances are also preserved. However, a key benefit we obtain is that we can *extend* the metric to the boundary of the manifold without introducing numerical instability as the metric at the boundary does not require us to divide by zero.

**Building conditional paths and vector fields on $\mathbb{S}_+^d$**. On any Riemannian manifold $\mathcal{M}^d$ that admits a probability density, it is possible to define a geodesic interpolant that connects two points between samples $x_0 \sim p_0$ to $x_1 \sim p_1$. A point traversing this interpolant, indexed by time $t \in [0, 1]$, can be expressed as $x_t = \exp_{x_0}(t \log_{x_0}(x_1))$. On general Riemannian manifolds, it is often not possible to obtain analytic expressions for the manifold exponential and logarithmic maps and as a result, traversing this interpolant requires the costly simulation of the Euler-Lagrange equations. Conveniently, under the Fisher-Rao metric $\mathring{\Delta}^d$ admits simple analytic expressions for the exponential and logarithmic maps—and consequently the geodesic interpolant. Moreover, due to the sphere-map $\varphi$ in eq. (6) the geodesic interpolant is also well-defined on $\mathbb{S}_+^d$. Such a result means that the conditional flow $x_t$ on $\mathbb{S}_+^d$ can be derived analytically from the well-known geodesics on a hypersphere, *i.e.,* they are the great circles but restricted to the positive orthant. Consequently, we may build all of the conditional flow machinery using well-studied geometric expressions for $\mathbb{S}_+^d$ in a numerically stable manner.

The target conditional vector field associated at $x_t$ can also be written in closed-form $u_t(x_t|x_0, x_1) = \log_{x_t}(x_1)/(1 - t)$ and computed exactly on $\mathbb{S}^d_+$. Intuitively, $u_t$ moves at constant velocity from $x_t$ in the direction of $x_1$ and presents a simple regression target to learn the vector field $v_\theta$. One practical benefit of learning conditional vector fields on $\mathbb{S}^d_+$ is that it allows for more flexible parameterisation of the vector field network $v_\theta$. Specifically, the network $v_\theta$ can be unconstrained and output directly in the ambient space $\mathbb{R}^{d+1}$ after which we can orthogonally project them to the tangent space of $x_t$. This is possible since we can take an *extrinsic* view on the geometry and isometrically embed $\mathbb{S}^d_+$ to the higher dimensional ambient space due to the Nash embedding theorem [35]. More formally, we have that $v_\theta(t, x_t) = \phi_{x_t}(\tilde{v}_\theta(t, x_t))$, where $\tilde{v}_\theta$ is the output in $\mathbb{R}^d$ and $\phi_{x_t} : \mathbb{R}^d \to \mathcal{T}_{x_t}\mathbb{S}^d_+$ and is defined as $\phi_{x_t}(\tilde{v}) = \tilde{v} - \langle x_t, \tilde{v} \rangle_2 x_t$.

In the absence of any knowledge we can choose an uninformative prior on $\mathbb{S}^d_+$ which is the uniform density over the manifold $p_1(x_1) = \sqrt{\det \mathbf{G}(x_1)}/\int_{\mathbb{S}^d_+} \sqrt{\det \mathbf{G}(x_1)}$, where $\mathbf{G}$ is the matrix representation of the Riemannian metric. However, a key asset of our construction, in contrast, to [23, 68], is that $p_1$ can be any source distribution since we operate on the *interpolant-level* by building geodesics between two points, $x_0, x_1 \in \mathbb{S}^d_+$. We now state the Riemannian CFM objective for $\mathbb{S}^d_+$:

$$\mathcal{L}_{\mathbb{S}^d_+}(\theta) = \mathbb{E}_{t, q(x_0, x_1), p_t(x_t|x_0, x_1)} \|v_\theta(t, x_t) - \log_{x_t}(x_1)/(1 - t)\|^2_{\mathbb{S}^d_+}, \quad t \sim \mathcal{U}(0, 1). \tag{7}$$

In a nutshell, our recipe for learning conditional flow matching for discrete data first maps the input data to $\mathbb{S}^d_+$. Then we learn to regress target conditional vector fields on $\mathbb{S}^d_+$ by performing Riemannian CFM which can be done easily as the hypersphere is a simple geometric object where geodesics can be stated explicitly. At inference, our flow pushes forward a prior on $p_1 \in \mathbb{S}^d_+$ to a desired target, $p_0$, which is then finally mapped back to $\mathring{\Delta}^d$ using $\varphi^{-1}$. A discrete category can then be chosen using any decoding strategy such as sampling using the mapped categorical or greedily by simply selecting the closest vertex of the probability simplex $\Delta^d$ to the final point at the end of inference.

### 3.3 The Fisher-Rao metric from Natural gradient descent

We now motivate the choice of the Fisher-Rao metric as not only a natural choice but also the optimal one on the probability simplex. We show that gradient descent of the general form $\delta\theta \mapsto \operatorname{argmin}_{|\delta\theta| \leq \epsilon} \mathcal{L}(\theta + \delta\theta)$ (for $\mathcal{L}(\theta) = \mathcal{L}(p_\theta)$ as in Eq. 7) converges to the gradient flow (of parameterised probabilities $p_\theta$, or of probability paths $p_{\theta, t}$) with respect to the Wasserstein distance on $\mathcal{P}(\mathcal{M})$ induced by Fisher-Rao metric $g_{\mathrm{FR}}$ over $\mathcal{M}$. Equivalently, we get the canonical metric over $\mathbb{S}^d_+$ due to the isometry. This presents a further justification for the use of the Fisher-Rao metric.

In order to present the gradient flow of $\mathcal{L} : \mathcal{P}(\mathcal{M}^d) \to \mathbb{R}$ in which $(\mathcal{M}^d, g)$ is a Riemannian manifold, we recall the basics of geometry over probability spaces [5, 76]. If $d_g$ is the geodesic distance associated to $g$ then $W_{2,g}$ will be the optimal transport distance over $\mathcal{P} = \mathcal{P}(\mathcal{M}^d)$ with cost $d_g^2(x, y)$. Then $(\mathcal{P}(\mathcal{M}^d), W_{2,g})$ is an infinite-dimensional Riemannian manifold, in which for $p \in \mathcal{P}(\mathcal{M}^d)$ we have the tangent space $\mathcal{T}_p\mathcal{P} \simeq \overline{\{\nabla_g \phi : \phi \in C^1_c(\mathcal{M}^d)\}}^{L^2_g(\mathcal{T}\mathcal{M}^d; p)}$, *i.e.*, the closure of gradient vector fields with respect to $L^2_g(\mathcal{T}\mathcal{M}^d; p)$-norm. This norm is defined by the Riemannian tensor $g^\mathcal{P}$ induced by $g$, which at $v, w \in \mathcal{T}_p\mathcal{P}$ is given by $g^\mathcal{P}(v, w) := \int_{\mathcal{M}^d} \langle v(x), w(x) \rangle_g \, dp(x)$. In particular, note that a choice of Riemannian metric $g$ over $\mathcal{M}^d$ specifies a unique metric $g^\mathcal{P}$ over $\mathcal{P}$.

In the following, at the onset, we assume a bounded metric, $g$, over $\Delta^d$, which we use only to state our Lipschitz dependence assumptions. If we compare categorical densities (elements of $\mathcal{M}^d = \mathcal{P}(\mathcal{A})$ via KL-divergence, then it is natural to compare distributions $\mu, \nu \in \mathcal{P}(\mathcal{M}^d)$ via the Wasserstein-like $W_{\mathrm{KL}}(\mu, \nu) := \min_{\pi \in \Pi(\mu, \nu)} \mathbb{E}_{(p_\omega, p_{\omega'}) \sim \pi}[\mathbb{D}_{\mathrm{KL}}(p_\omega \| p_{\omega'})]$. In the next proposition, we show that the Fisher-Rao metric appears naturally in the continuum limit of our gradient descent over $\mathcal{P}(\mathcal{M}^d)$.

**Proposition 1.** *Assume that there exists a bounded Riemannian metric $g$ over $\Delta^d$ such that the parameterisation map $\theta \mapsto p = p(\theta)$ is Lipschitz and differentiable from $\Theta$ to $(\mathcal{P}(\mathcal{M}), W_{2,g})$. Then the "natural gradient" descent of the form:*

$$p(\theta_{n+1}) \in \operatorname{argmin}\{\mathcal{L}(p(\theta_{n+1})) : W_{\mathrm{KL}}(p(\theta_{n+1}), p(\theta_n)) \leq \epsilon\} \tag{8}$$

*approximates, as $\epsilon \to 0^+$, the gradient flow of $\mathcal{L}$ on manifold $(\mathcal{P}(\mathcal{M}^d), W_{g_{\mathrm{FR}},2})$ with metric $g_{\mathrm{FR}}^{\mathcal{P}}$ induced by Fisher-Rao metric $g_{\mathrm{FR}}$:*

$$\frac{\mathrm{d}}{\mathrm{d}s} p(\theta(s)) = \nabla_{g_{\mathrm{FR}}^{\mathcal{P}}} \mathcal{L}(p(\theta(s))). \tag{9}$$

For the proof, see §C. We distinguish the results of Proposition 1 from those of Natural Gradients used in classical NN optimisation such as KFAC [52]. Note that in regular NN training, Natural Gradients [58] implement a second-order optimisation to tame the gradient descent, at a nontrivial computational cost. Thus, the above proposition implies that, just by selecting $g_{\mathrm{FR}}$ metric over $\Delta^d$, we directly get the benefits that are equivalent to the regularisation procedure of Natural Gradient.

## 3.4 FISHER-FLOW Matching with Riemannian optimal transport

We now demonstrate how to build conditional flows that minimise a Riemannian optimal transport (OT) cost. Flows constructed by following an optimal transport plan enjoy several theoretical and practical benefits: 1. They lead to shorter global paths. 2. No two paths cross which leads to lower variance gradients during training. 3. Paths that follow the transport plan have lower kinetic energy which often leads to improved empirical performance due to improved training dynamics [64].

The Riemannian optimal transport for FISHER-FLOW can be stated for either $\mathring{\Delta}^d$ under the Fisher-Rao metric or $\mathbb{S}_+^d$. Both instantiations lead to the same optimal plan due to the isometry between the two manifolds. Specifically, we couple $q(x_0), q(x_1)$ via the Optimal Transport (OT) plan $\pi(x_0, x_1)$ under square-distance cost $c(x, y) := d^2(x, y)$—*i.e.,* $\pi(x_0, x_1)$ will be the minimiser of $\mathbb{E}_{(x_0, x_1) \sim \pi'}[d^2(x_0, x_1)]$ amongst all couplings $\pi'$ of fixed marginals $q(x_0), q(x_1)$. Now, recall that Wasserstein distance $W_2$ over $\mathcal{P}(\mathbb{S}_+^d)$ is defined as $W_2(\mu, \nu) := \min_{\pi'} \mathbb{E}_{(x,y) \sim \pi'}[d_{\mathbb{S}_+^d}^2(x, y)]$, in which the minimisation is amongst transport plans from $\mu$ to $\nu$, defined as probability measures over $\mathbb{S}_+^d \times \mathbb{S}_+^d$ whose two marginals are respectively $\mu$ and $\nu$ [75]. Since $\mathbb{S}_+^d$ is a smooth bounded uniquely geodesic Riemannian manifold with boundary, the metric space $(\mathcal{P}(\mathbb{S}_+^d), W_2)$ is uniquely geodesic and we have the following "informal" proposition (see §D for the full statement):

**Proposition 2.** *For any two Borel probability measures $p_0, p_1 \in \mathcal{P}(\mathbb{S}_+^d)$, there exists a unique OT-plan $\pi$ between $p_0, p_1$. If $e_t(x_0, x_1)$ is the constant-speed parameterisation of the unique geodesic of extremes $x_0$ and $x_1$, and $e_t : \mathbb{S}_+^d \times \mathbb{S}_+^d \to \mathbb{S}_+^d$ is given by $e_t(x_0, x_1) := \exp_{x_0}(t \log_{x_0}(x_1))$, then there exists a unique Wasserstein geodesic $(p_t)_{t \in [0,1]}$ connecting $p_0$ to $p_1$, given by*

$$p_t := (e_t)_{\#} \pi \in \mathcal{P}(\mathbb{S}_+^d), \quad t \in [0, 1]. \tag{10}$$

The complete statement of Proposition 4 along with its proof is provided in §D. As a consequence of Proposition 2 we use the Wasserstein geodesic as our target conditional probability path. Operationally, this requires us to sample from marginals $x_0 \sim p_0$ and $x_1 \sim p_1$ and solve for the OT plan $\pi$ using the squared distance on $\mathbb{S}_+^d$ as the cost which is done using the Sinkhorn algorithm [32].

## 3.5 Training FISHER-FLOW

**Generalising to sequences**. Many problems in generative modeling over discrete data are concerned with handling a set or a sequence of discrete objects. For complete generality, we now extend FISHER-FLOW to a sequence of discrete data by modeling it as a Cartesian product of categorical distributions. Formally, for a sequence of length $k$ we have a distribution over a product manifold $\mathcal{P}(\boldsymbol{\Delta}) := \mathcal{P}(\Delta_1^d) \times \cdots \times \mathcal{P}(\Delta_k^d)$. Equipping each manifold in the product with the Fisher-Rao metric allows us to extend the metric in a natural way to $\boldsymbol{\Delta}$. Moreover, by invoking the diffeomorphism using the sphere-map $\varphi$ independently we achieve the product of $d$-hyperspheres restricted to the positive orthant. Stated explicitly, a sequence of categorical distributions is $\mathcal{P}(\mathbf{S}_+) := \mathcal{P}((\mathbb{S}_+^d)_1) \times \cdots \times \mathcal{P}((\mathbb{S}_+^d)_k)$. Due to the factorisation of the metric across the product space, we can build independent flows on each manifold $\mathbb{S}_+^d$ and couple them in a natural way using the product metric to induce a flow on $\mathbf{S}_+$.

**Training**. We detail our method for training FISHER-FLOW in Algorithm 1 in §F.2. Training FISHER-FLOW requires two input distributions: a source and a target one. In the case of unconditional generation, one can take $p_0 = \mathcal{U}(\mathbb{S}_+^d)$, by default. In some settings, it is possible to incorporate

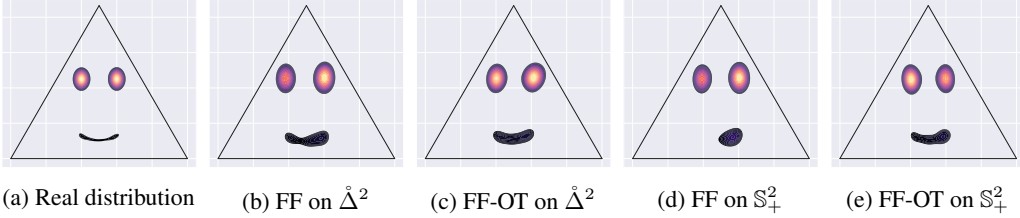

(a) Real distribution     (b) FF on $\mathring{\Delta}^2$     (c) FF-OT on $\mathring{\Delta}^2$     (d) FF on $\mathbb{S}^2_+$     (e) FF-OT on $\mathbb{S}^2_+$

Figure 2: Synthetic experiments on learning a distribution resembling a smiley face on $\mathring{\Delta}^2$.

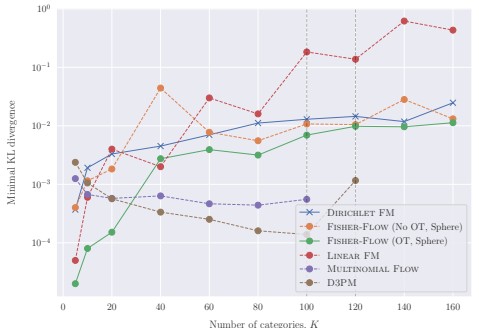
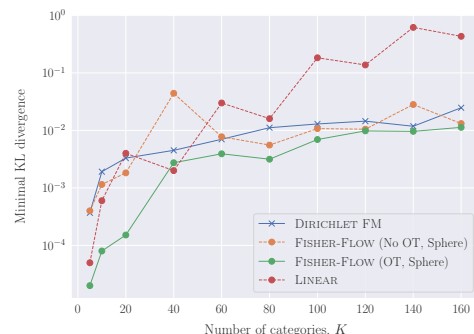

(a) Results of the ablation study. Missing points for Multinomial Flow [41] and D3PM [8] are `NaNs`.

(b) KL divergence of FISHER-FLOW against DIRICHLET FM.

Figure 3: Toy experiment from Stark et al. [68]. Minimal KL divergence over 5 seeds is reported.

additional conditional information, $c$. This can easily be accommodated in FISHER-FLOW by directly inputting this into the parameterised vector field network with $v_\theta(\cdot)$ becoming $v_\theta(\cdot|c)$.

## 4 Experiments

We now investigate the empirical caliber of FISHER-FLOW on a range of synthetic and real-world benchmarks outlined below. Unless stated otherwise, all instantiations of FISHER-FLOW use the optimal transport coupling on minibatches. Exact implementation details are included in §F.

### 4.1 Synthetic experiments

**Density estimation**. In this first experiment, we model an empirical categorical distribution visualized on $\mathring{\Delta}^2$. In Figure 2, we observe that FISHER-FLOW instantiated on $\mathbb{S}^2_+$ with OT is the best in modeling the ground truth distribution. Both learning on the simplex and the positive orthant benefit from OT.

**Density learning in arbitrary dimensions**. We also consider the toy experiment of Stark et al. [68], where we seek to model a random distribution over $(\Delta^K)^4$ for $K \in \mathbb{N}^\star$. The KL divergence between the estimated distribution over 512,000 samples and the true generated distribution is used as the evaluation metric. Details are provided in §F.4.1. Results in Figure 3b demonstrate that FISHER-FLOW outperforms DIRICHLET FM, while remaining competitive against D3PM [8] and Multinomial Flow [41], especially in high dimensions, in which both exhibit unstable behaviour. We also conduct an ablation in Figure 3a and find that using optimal transport helps for both FISHER-FLOW on $\mathring{\Delta}^d$ and $\mathbb{S}^d_+$, with the latter leading to the best performance.

### 4.2 Promoter DNA sequence design

We assess the ability of FISHER-FLOW to generate DNA sequences. Promoters are DNA sequences that determine where on a gene DNA is transcribed into RNA; they contribute to determining how much transcription happens [36]. The goal of this task is to generate promoter DNA sequences conditioned on a desired transcription signal profile. Solving this problem would enable one to control the expression level of any synthetic gene, *e.g.,* in the production of antibodies. For a detailed dataset background, see §F.1 in Avdeyev et al. [9].

Table 1: MSE of the transcription profile conditioned on generated promoter DNA sequences over the test set. The last 3 MSE and PPL values are from 5 independent experiments. The remaining numbers are taken directly from Stark et al. [68].

| Model | MSE ($\downarrow$) | PPL ($\downarrow$) |
|---|---|---|
| BIT DIFFUSION (BIT-ENCODING) | 0.041 | — |
| BIT DIFFUSION (ONE-HOT ENCODING) | 0.040 | — |
| D3PM-UNIFORM | 0.038 | — |
| DDSM | 0.033 | — |
| LANGUAGE MODEL | $0.034 \pm 0.001$ | $2.247 \pm 0.102$ |
| DIRICHLET FM | $0.034 \pm 0.004$ | $\mathbf{1.978 \pm 0.006}$ |
| FISHER-FLOW (ours) | $\mathbf{0.029 \pm 0.001}$ | $1.4 \pm 2.7$ |

Table 2: Perplexities (PPL) values for different methods for enhancer DNA generation. Lower PPL is better. Values are an average and standard error over 5 seeds.

| Method | Melanoma PPL ($\downarrow$) | Fly Brain PPL ($\downarrow$) |
|---|---|---|
| Random Sequence | 895.88 | 895.88 |
| Language Model | $2.22 \pm 0.09$ | $2.19 \pm 0.10$ |
| DIRICHLET FM | $2.25 \pm 0.01$ | $2.25 \pm 0.02$ |
| FISHER-FLOW (ours) | $\mathbf{1.4 \pm 0.1}$ | $\mathbf{1.4 \pm 0.66}$ |

Table 3: Results on QM9. Higher is better. The baseline numbers are taken from the cited papers. The numbers reported for FlowMol are those for the uniform distribution and end-point parameterisation. Our numbers are for 1,000 molecules.

| Method | Atoms S (%) | Mols Val (%) | Mols. S (%) |
|---|---|---|---|
| FISHER-FLOW (ours) | 98.6 | 95.3 | 88.2 |
| JODO [44] | 99.4 | 98.9 | 98.7 |
| EquiFM [67] | 99.4 | 94.4 | 93.2 |
| FlowMol [29] | 98.9 | 96.9 | 84.2 |

Figure 4: Generated molecules using FISHER-FLOW on QM9.

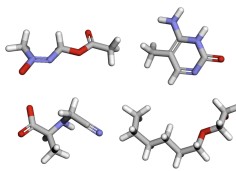

**Results**. Our experimental evaluation closely follows prior work [9, 68]. We report the MSE between the signal of our conditionally generated sequence and the target one, a human genome promoter sequence (MSE in Table 1), both given by the same pre-trained Sei model [25]. We train our model on 88,470 promoter sequences, each of length 1,024, from a database of human promoters [40], each sequence having an associated expression level indicating the likelihood of transcription at each DNA position. As shown in Table 1, FISHER-FLOW outperforms baseline methods DDSM [9] and DIRICHLET FM [68] on the MSE evaluation. Perplexities (PPL) from FISHER-FLOW on the test set are also better than the baselines and, on average, improve over DIRICHLET FM.

## 4.3 Enhancer DNA design

Enhancers are DNA sequences that regulate the transcription of DNA in specific cell types (*e.g.,* melanoma cells). Prior work has made use of generative models for designing enhancer DNA sequences in specific cells [71]. Following Stark et al. [68], we report the perplexity over sequences as the main measure of performance. We also include results on Fréchet Biological Distance (FBD) with pre-trained classifiers provided in DIRICHLET FM [68], cf. §F.4.3. Nevertheless, those classifiers perform poorly on cell-type classification of Enhancer sequences, with test set accuracies of 11.5% and 11.2% on the Melanoma and FlyBrain datasets, respectively; thus, metrics derived from these are not representative of model quality, which we still included in §F.4.3 for transparency.

**Results**. We report our results in Table 2, with FBD reported in Table 5. We observe that FISHER-FLOW obtains significantly better performance than DIRICHLET FM, which highlights its ability to fit the distribution of Melanoma and FlyBrain DNA enhancer sequences. Moreover, we also note that our method improves over the language model baseline on both datasets, which bolsters the belief that FISHER-FLOW can be used in similar settings to those of autoregressive models.

## 4.4 *De novo* molecule generation

In this experiment, we evaluate FISHER-FLOW's ability to generate molecules unconditionally, a.k.a. "*de novo*". The difficulty in this task is that we are interested in generating the positions of the molecules, their atom types, their charges, and the bonds between these, resulting in a high dimensional space with both discrete and continuous data $(\mathbb{R}^d)^n \times (\Delta^a)^n \times (\Delta^c)^n \times (\Delta^e)^{n^2}$, where $n \in \mathbb{N}^\star$ is the number of atoms, $a$ possible atom types, $c$ charges, and $e$ bonds. We train our model over the QM9 dataset [61, 60]. We report the percentage of stable atoms within molecules, valid molecules, and stable molecules. Our implementation is mostly based on that of [29].

**Results**. We report our results in Table 3. We also provide some qualitative examples in Figure 4. As we can see, FISHER-FLOW compares well on all metrics to SIMPLEX-FLOW on all metrics. Nonetheless, it must be reported that the latter, trained with a Gaussian prior, endpoint parameterisation and cosine time schedule performed substantially better than both flow-based methods, closing the gap with the other baselines. It is likely that a more extensive exploration of priors, time parameterisations and other hyperparameters would increase FISHER-FLOW's performance.

### 4.5 Language modelling

Finally, we test the language modelling capabilities of FISHER-FLOW. To do so, we train the model on the LM1B dataset [24], a large language modelling dataset containing about 800,000 words. For this experiment, we extend FISHER-FLOW to a masked path as is done by [62, 65]: we define the probability path as $p_t = \kappa_t p_M + (1 - \kappa_t) p_{\text{unif}}$, where $\kappa : [0, 1] \to [0, 1]$ is a noise scheduler. Here, $p_M$ is the Fisher-Rao geodesic between the target, $x_0$, and the designated mask token $M$, while $p_{\text{unif}}$ is also a Fisher-Rao geodesic between a sample from a uniform distribution and $x_0$. It is thus a convex combination of probability paths. Using a denoising architecture enables us to rewrite the

Table 4: Test perplexities on the LM1B dataset. All baselines are taken from concurrent work MDLM by Sahoo et al. [62]. Best diffusion or flow-matching method is in bold font.

| | Method | Parameters | PPL ($\downarrow$) |
|---|---|---|---|
| Diffusion | BERT-MOUTH | 110M | $\leq 142.89$ |
| | D3PM (ABSORB) | 70M | $\leq 77.50$ |
| | DIFFUSION-LM | 80M | $\leq 118.62$ |
| | DIFFUSIONBERT | 110M | $\leq 63.78$ |
| | SEDD (33B TOKENS) | 110M | $\leq 32.79$ |
| AR | TRANSFORMER (33B TOKENS) | 110M | 22.32 |
| | TRANSFORMER (327B TOKENS) | 110M | 20.86 |
| DM/FM | MDLM (33B TOKENS) | 110M | $\leq 27.04$ |
| | FISHER-FLOW (33B TOKENS) (ours) | 110M | $\leq \mathbf{26.51}$ |
| | MDLM (327B TOKENS) | 110M | $\leq 23.00$ |
| | FISHER-FLOW (327B TOKENS) (ours) | 110M | $\leq \mathbf{22.42}$ |

original loss as a weighted negative log likelihood $-\mathbb{E}[\log p(x_0 \mid x_t)]$. This allows us to calculate an upper bound on the test perplexity, a natural evaluation metric for language modelling [62, 65].

**Results**. The results are given in Table 4. As one can observe, using the Fisher-Rao metric enables better performance than MDLM. Yet, the gap with auto-regressive methods is still significant.

## 5 Related work

**Geometric generative models**. There are several methods for defining generative models over Riemannian manifolds, the most pertinent to this work include diffusion models [43, 28], normalising flows [20, 53, 15, 26]. For molecular tasks that require generating nodes and edges, equivariant variants of diffusion and flow-based models are a natural choice [42, 78].

**Discrete diffusion and flow models**. Discrete generative models diffusion and flow models can be categorised into either relaxations to continuous spaces [47, 27], or methods that use continuous-time Markov chains with sophisticated transition kernels [8, 80, 23, 50], with some matching autoregressive models [34]. Defining discrete data on the simplex has also been explored in the context of generative models [37, 51, 68]. FISHER-FLOW is fundamentally different from existing works [8, 23, 2, 50] in that we consider a continuous relaxation of the discrete space and construct vector fields on $\mathbb{S}_+^d$. Finally, concurrent to our work Dunn and Koes [29] propose simplex flow matching, and Boll et al. [17] introduced $e$-geodesic flows that leverage the Fisher-Rao metric on the assignment manifold [17]. Simplex flow-matching differs from FISHER-FLOW in that it does not make use of the Fisher-Rao metric. We include a detailed comparison between FISHER-FLOW in relation to DFM and $e$-Geodesic Flow Matching [17] in §E.1.

## 6 Conclusion

In this paper, we introduce FISHER-FLOW a novel generative model for discrete data. Our approach offers a novel perspective and reparameterises discrete data to live on the positive orthant of a $d$-hypersphere, which allows us to learn categorical densities by performing Riemannian flow matching. Empirically, FISHER-FLOW improves performance on synthetic and biological sequence design tasks over comparative discrete diffusion and flow matching models while being more general as a framework. While FISHER-FLOW enjoys favorable theoretical properties with strong empirical performance, our method is not fully developed for language modeling domains. Consequently, a natural direction for future work is to design variations of FISHER-FLOW capable of handling larger sequence lengths and discrete categories as found in language domains.

## Acknowledgements

We thank Alexander Tong for his generous time, help, and guidance in helping with the language modeling experiments. OD is supported by both Project CETI and Intel. MP is supported by CenIA and by Chilean Fondecyt grant n. 1210426. AJB is partially supported by an NSERC Post-doc fellowship. This research is partially supported by EPSRC Turing AI World-Leading Research Fellowship No. EP/X040062/1 and EPSRC AI Hub on Mathematical Foundations of Intelligence: An "Erlangen Programme" for AI No. EP/Y028872/1

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

# A   Broader Impacts

We would like to emphasise that our paper is mainly theoretical and establishes generative modeling of discrete data using flow matching by continuously reparameterising points onto a statistical manifold equipped with the Fisher-Rao metric. However, more broadly discrete generative modeling based on diffusion models and flow matching has important implications in various fields. In biology, these models enable the generation of novel biological sequences, facilitating the development of new therapeutics. However, the same technology poses risks if exploited maliciously, as it could be used to design harmful substances or biological weapons. In language modeling, the capability to generate coherent and contextually relevant text can significantly enhance productivity, creativity, and communication. Nevertheless, the advent of superhuman intelligence through advanced language models raises concerns about potential misuse, loss of human control, and ethical dilemmas, highlighting the need for robust oversight and ethical guidelines.

# B   Geometry of the Simplex

We introduce here very briefly properties of geometry on the simplex that we use in this paper. Our main reference for these results is Åström et al. [81]. Note that our implementation for most of these properties relies on that of Axen et al. [10], which we port to Python. Recall that a $d$-simplex, for $d \in \mathbb{N}^\star$, is defined as $\Delta^d := \{x \in \mathbb{R}^{d+1} | \mathbf{1}^\top x = 1, x \geq 0\}$. When equipped with the Fisher-Rao metric, it becomes a Riemannian manifold that is isometric to the positive orthant of the $d$-sphere of in $\mathbb{R}^{d+1}$. That is to say, $\psi : \Delta^d \to \mathbb{S}_+^d, (x_0, \ldots, x_d) \mapsto (2\sqrt{x_0}, \ldots, 2\sqrt{x_d})$ is a diffeomorphism, where $\mathbb{S}_+^d := \{x \in \mathbb{R}^{d+1} : \|x\|_2 = 2, x \geq 0\}$; we call $\psi$ the "sphere-map".

In the following, $\mathring{\Delta}^d$ denotes the interior of the simplex, and $\mathcal{T}_p\Delta^d := \{x \in \mathbb{R}^{d+1} : \mathbf{1}^\top x = 0\}$ the tangent space at point $p$. The exp map on the simplex is given by, for all $p \in \mathring{\Delta}^d$, $v \in \mathcal{T}_p\Delta^d$,

$$\exp_p(v) = \frac{1}{2}\left(p + \frac{v_p^2}{\|v_p^2\|^2}\right) + \frac{1}{2}\left(p - \frac{v_p^2}{\|v_p^2\|^2}\right)\cos(\|v_p\|) + \frac{\sqrt{p}}{\|v_p\|}\sin(\|v_p\|), \qquad (11)$$

where $v_p := \frac{v}{\sqrt{p}}$, and squares, square roots and quotients of vectors are meant element-wise. Similarly, the log map is given by, for $p, q \in \mathring{\Delta}^d$,

$$\log_{x_0}(x_1) = \frac{d_{\Delta^d}(p, q)}{\sqrt{1 - \langle\sqrt{p}, \sqrt{q}\rangle}}\left(\sqrt{pq} - \langle\sqrt{p}, \sqrt{q}\rangle p\right), \qquad (12)$$

where the product is meant element-wise, and the distance is

$$d_{\Delta^d} = 2\arccos(\langle\sqrt{p}, \sqrt{q}\rangle). \qquad (13)$$

The Riemannian metric at point $p \in \mathring{\Delta}^d$ for vectors $u, v \in \mathcal{T}\Delta^d$ is given by

$$\langle u, v\rangle_p = \left\langle \frac{u}{\sqrt{p}}, \frac{v}{\sqrt{p}}\right\rangle. \qquad (14)$$

Finally, for parallel transport, we use the sphere-map, perform parallel-transport on the sphere, and invert the sphere-map.

The relevance of the Fisher-Rao metric stems from the following two characterisations:

- *The Fisher-Rao metric is the leading-order approximation of the Kullback-Leibler divergence [4, 11].* Recall the general setting: if a $d$-dimensional manifold of probability densities $\mathcal{M}^d$ is parameterised by a differentiable map $\theta \mapsto p_\theta$ from a submanifold $\Theta \subseteq \mathbb{R}^D$ (note that the requirement $D = d$ is not necessary for the following computations to make sense), then for fixed $\theta_0 \in \Theta$ we may Taylor-expand

$$p(\theta) = p(\theta_0) + \sum_{j=1}^{D}(\theta^j - \theta_0^j)\frac{\partial p(\theta_0)}{\partial\theta^j} + o(|\theta - \theta_0|),$$

and a straightforward computation gives

$$D_{KL}(p(\theta_0)||p(\theta)) \quad = \quad \frac{1}{2}\sum_{j,k=1}^{D}(\theta^j - \theta_0^j)(\theta^k - \theta_0^k)\,\mathbb{E}_{p(\theta)}\left[\frac{\partial \log p}{\partial \theta^j}\frac{\partial \log p}{\partial \theta^k}\right]\Bigg|_{\theta=\theta_0} + o(|\theta - \theta_0|^2)$$

$$:= \quad \frac{1}{2}\sum_{j,k=1}^{D} g_{jk}(\theta_0)[\theta^j - \theta_0^j, \theta^k - \theta_0^k] + o(|\theta - \theta_0|^2).$$

Thus the matrix $g(\theta_0) = (g_{ij}(\theta_0))_{i,j=1}^{D}$ defines the quadratic form on the tangent space $\mathcal{T}_{\theta_0}\Theta$ which best approximates $D_{KL}(p(\theta_0)||p(\theta))$ in the limit $\theta \to \theta_0$. In the coordinates $\Theta = \Delta^d \subset \mathbb{R}^{d+1}$, when we parameterise probabilities over $K = d+1$ classes numbered $0, \ldots, d$ via the "tautological" parameterisation $\theta = p$ for $p \in \Delta^d$, (explicitly, in this parameterisation class $i$ has probability $p_\theta(i) = \theta^i = p^i$), then we obtain $\frac{\partial \log p_\theta}{\partial \theta^j} = \frac{1}{p^j}\delta(i = j)$ and

$$g_{jk}(p) = \mathbb{E}_{p(\theta)}\left[\frac{\partial \log p}{\partial \theta^j}\frac{\partial \log p}{\partial \theta^k}\right] = \sum_{i=1}^{d+1} p^i \frac{1}{p^j}\delta(i = j)\frac{1}{p^k}\delta(i = k) = \frac{1}{p^j}\delta(j = k).$$

Thus $g(p)[u, v] = g_{\mathrm{FR}}(p)[u, v] = \sum_{i=1}^{d+1}\frac{u^i v^i}{p^i}$ as before.

- *The Fisher-Rao metric is up to rescaling, the only metric that is preserved under sufficient statistics.* First, for $2 \leq K' \leq K$, a define a map $M : \mathcal{P}([K']) \to \mathcal{P}([K])$ to be a *Markov map* if there exist probability measures $q_1, \ldots, q_{K'} \in \mathcal{P}([K])$ such that for $p \in \mathcal{P}([K'])$ we have $M(p) = \sum_{k=1}^{K'} p(k)q_k$. In other words, representing probability spaces as simplices and denoting $d = K - 1, d' = K' - 1$, we have that $M$ is a Markov map if the simplex $\Delta^{d'}$ is affinely mapped under $M$ to a $d'$-dimensional simplex in $\Delta^d$ (the vertices of the image simplex have been denoted above by $q_1, \ldots, q_{K'}$).

  Then a restatement of Chentsov's theorem [74, Thm. 11.1], [11, Thm. 1.2] is that if a sequence of Riemannian metrics $g_d$ over $\Delta^d$ defined for $d \geq 2$ satisfies the property that for any $1 \leq d' \leq d$ any Markov morphism $M : \Delta^{d'} \to \Delta^d$ is an isometry with respect to metrics $g_{d'}, g_d$, then there exists $C > 0$ such that each of the $g_d$ is $C$ times the Fisher-Rao metric on $\Delta^d$.

  A common reformulation, interpreting the Markov map reparameterisations $M : \mathcal{P}([K']) \to \mathcal{P}([K])$ of $\mathcal{P}([K'])$ as sufficient statistics, is to say that Fisher-Rao metrics are (up to a common rescaling for all $d$) the only metrics that are invariant under mapping probability measures to sufficient statistics.

## C   Details and proofs for Section 3.3

We here recall the setup: we are considering a loss function $\mathcal{L} : \mathcal{P}(\mathcal{M}^d) \to \mathbb{R}$, in which $\mathcal{M}^d$ is a Riemannian manifold, specifically it will be the simplex $\Delta^d$ endowed with a Riemannian metric $g$. Points $p_\omega \in \mathcal{M}^d$ represent categorical distributions, as $\mathcal{M}^d$ was obtained from $\mathcal{P}(\mathcal{A})$ by parametrising it with the simplex $\Delta^d$, thus inducing a differentiable structure.

The space $\mathcal{P}(\mathcal{M}^d)$ is then endowed with the Wasserstein distance $W_{2,g}$ induced by the Riemannian geodesic distance of $(\mathcal{M}^d, g)$. Then $\mathcal{P} = (\mathcal{P}(\mathcal{M}^d), W_{2,g})$ can be given a Riemannian metric structure too, defined as follows [5, 76]. For $p \in \mathcal{P}$ the tangent space $\mathcal{T}_p\mathcal{P}$ is identified with the $L^2(p; g)$-closure of the space of vector fields $v : \mathcal{M}^d \to \mathcal{T}\mathcal{M}^d$ which are the gradient of a $C_c^1$-function $\psi : \mathcal{M}^d \to \mathbb{R}$. Here we have for $v = \nabla\psi \in \mathcal{T}_p\mathcal{P}$

$$\|v\|_{L^2(p;g)}^2 := \int_{\mathcal{M}^d}\|v(p_\omega)\|_g^2 dp(p_\omega),$$

and the corresponding Riemannian tensor induced by $g$ over $v, w \in \mathcal{T}_p\mathcal{P}$ is given by

$$g^{\mathcal{P}}(v, w) := \int_{\mathcal{M}^d}\langle v(p_\omega), w(p_\omega)\rangle_g dp(p_\omega).$$

For further details see Villani [76, Ch. 13].

In order to find a well-behaved metric over $\mathcal{M}^d$, we start by considering $\mathcal{M}^d$ (which in our case is the statistical manifold parameterising the space of categorical probabilities $\mathcal{P}(\mathcal{A})$) with the KL-divergence as a comparison tool for its elements. We will use this divergence in order to regularise

the gradient descent of a loss function $\mathcal{L} : \mathcal{P}(\mathcal{M}^d) \to \mathbb{R}$, and to do so we introduce the KL-optimum coupling which for $\mu, \nu \in \mathcal{P}(\mathcal{M}^d)$ takes the value

$$W_{\mathrm{KL}}y(\mu, \nu) := \min \left\{ \mathbb{E}_{(p_\omega, p_{\omega'}) \sim \pi} \left[ \mathbb{D}_{\mathrm{KL}}(p_\omega \| p_{\omega'}) \right] : \pi \in \mathcal{P}(\mathcal{M}^d \times \mathcal{M}^d) \text{ has marginals } \mu, \nu \right\}.$$

In words, $W_{\mathrm{KL}}$ determines the smallest average displacement required for moving $\mu$ to $\nu$, in which displacements between elements of $\mathcal{M}^d \simeq \mathcal{P}(\mathcal{A})$ are quantified by $\mathbb{D}_{\mathrm{KL}}$-divergence.

We then use this distance to regularise the gradient descent of $\mathcal{L}$, and show that then the gradient descent converges to the Wasserstein gradient flow on $\mathcal{L}$, for precisely the Wasserstein distance $W_{2, g_{\mathrm{FR}}}$ induced by the $g_{\mathrm{FR}}$-metric over $\mathcal{M}^d$.

Here we consider a Riemannian metric structure $g$ on $\mathcal{M}^d$, which we assume to be bounded on the interior $\mathring{\Delta}^d$, *i.e.,* to have bounded coefficients when expressed in the parametrisation, which is only used in order to give a rough Lipschitz hypothesis on the underlying parametrisations.

> **Proposition 3** (extended version of Proposition 1). *Assume that $g$ is a bounded Riemannian metric over $\Delta^d$ such that the parametrisation map $\theta \mapsto p = p(\theta) : \Theta \to (\mathcal{P}(\mathcal{M}^d), W_{2,g})$ is Lipschitz and differentiable . Then the "natural gradient" descent of the form:*
>
> $$p(\theta_{n+1}) \in \mathrm{argmin} \left\{ \mathcal{L}(p(\theta_{n+1})) : W_{\mathrm{KL}}(p(\theta_{n+1}), p(\theta_n)) \leq \epsilon \right\} \tag{15}$$
>
> *approximates, as $\epsilon \to 0^+$, the gradient flow of $\mathcal{L}$ on manifold $(\mathcal{P}(\mathcal{M}^d), W_{g_{\mathrm{FR}}, 2})$ with metric $g_{\mathrm{FR}}^{\mathcal{P}}$ induced by Fisher-Rao metric $g_{\mathrm{FR}}$:*
>
> $$\frac{\mathrm{d}}{\mathrm{d}s} p(\theta(s)) = \nabla_{g_{\mathrm{FR}}^{\mathcal{P}}} \mathcal{L}(p(\theta(s))). \tag{16}$$

*Proof.* We restrict the discussion to the case that $p(\theta)$ is supported in the region $\Delta_c^d := \{x \in \mathbb{R}^d : \mathbb{1} \cdot x = 1, \ x_i \geq c, 1 \leq i \leq d\}$, and the general result can be recovered by taking $c \to 0^+$. Restricted to this set, it is easy to verify that $\mathbb{D}_{\mathrm{KL}}$ is bounded.

**Step 1.** Note that by a small modification of the proof, we can apply Villani [76, Thm. 10.42] to $\Delta^d$ with cost equal to $\mathbb{D}_{\mathrm{KL}}$, and obtain that the $W_{\mathrm{KL}}$-distance between an admissible competitor $p(\theta + \delta\theta)$ in Eq. 15 and $p(\theta_n)$ is realised by a transport plan $T^{\delta\theta}$, such that we have $p(\theta + \delta\theta) = T_\#^{\delta\theta} p(\theta)$. By definition of $W_{\mathrm{KL}}$ and due to Chebyshev's inequality, for all $C > 0$, the set of points $S_C$ that $T^{\delta\theta}$ moves by more than $C\epsilon$ in $\mathbb{D}_{\mathrm{KL}}$-distance has $p(\theta)$-measure not larger than $1/C$. Furthermore, $T^{\delta\theta}$ is uniformly bounded over $\Delta_c^d \setminus S_C$ by our initial hypothesis. By approximating this transport plan by a flow (one can adapt the ideas from *e.g.,* Santambrogio [63, Thm. 4.4] for this contstruction) over $S_C$, we can find a vector field $v^{\delta\theta}$ such that $v^{\delta\theta}(p_\omega) = \frac{1}{\epsilon} \log_{p_\omega}(T^{\delta\theta}(p_\omega)) + o_\epsilon(|\delta\theta|)$ for $p_\omega \in S_C$, with error uniformly bounded in $p_\omega \in \mathcal{M}$. We then extend $v^{\delta\theta}$ arbitrarily outside $S_C$. This procedure associates to each small enough change $\delta\theta$ a vector field $v_{\delta\theta} \in T_{p(\theta)}\mathcal{P}$ which whose time-$\epsilon$ flow, denoted $\phi_{v_{\delta\theta}}(t = \epsilon, \cdot)$ pushes measure $p(\theta)$ to a measure approaching $p(\theta + \delta\theta)$ in the limit $\epsilon \to 0, C \to \infty$.

**Step 2.** We approximate the optimisation problem Eq. 15. For the constraint, we recall that as noted in Appendix B, we have Taylor expansion $\mathbb{D}_{\mathrm{KL}}(p_\omega \| p_{\omega'}) = \frac{1}{2} \|\omega - \omega'\|_{g_{\mathrm{FR}}}^2 + O(\|\omega - \omega'\|^3)$. For approximating $\mathcal{L}$ we use its differentiability and get $\mathcal{L}(p(\theta')) = \mathcal{L}(p(\theta)) + d\mathcal{L}(p(\theta))[v]$, for $v \in \mathcal{T}_p\mathcal{P}$. Thus minimisation problem Eq. 15 is well approximated, (in the limits mentioned in the previous step) by

$$p(\theta_{n+1}) = (\phi_{v_{\delta\theta}}(1, \cdot))_\# \, p(\theta_n), \quad v_{\delta\theta} \in \mathrm{argmin}_v \left( \epsilon \, d\mathcal{L}(p(\theta))[v] : \langle v, v \rangle_{g_{\mathrm{FR}}^{\mathcal{P}}} = 1 \right), \tag{17}$$

in which we used a rescaling compared to previous step, given by $v \mapsto \epsilon v$. This means that we used the associated flow up to time 1 rather than time $\epsilon$, and thus the minimisation has to be taken amongst elements $v \in T_{p(\theta)}\mathcal{P}$ and we approximate the constraint by $\langle v, v \rangle_{g_{\mathrm{FR}}^{\mathcal{P}}} = 1$, which replaces the correct constraint $W_{\mathrm{KL}}(p(\theta + \delta\theta), p(\theta)) = \epsilon$.

**Step 3.** In the optimisation Eq. 17, we have a quadratic constraint over the vector space $T_{p(\theta_n)}\mathcal{P}$, and thus we can use Lagrange multipliers, and for the optimiser we need to look for critical points of $v \mapsto \epsilon d\mathcal{L}(p(\theta))[v] + \frac{\lambda}{2} \langle v, v \rangle_{g_{\mathrm{FR}}^{\mathcal{P}}}$, in which $\lambda$ is the Lagrange multiplier, to be fixed at the end using the constraint. This gives the following characterisation of the optimiser $v_{\delta\theta}^*$:

$$\forall w \in T_{p(\theta)}\mathcal{P}, \quad \langle v_{\delta\theta}^*, w \rangle_{g_{\mathrm{FR}}^{\mathcal{P}}} = -\frac{\lambda}{\epsilon} d\mathcal{L}(p(\theta))[w] \iff v_{\delta\theta}^* = -\frac{\lambda}{\epsilon} \nabla_{g_{\mathrm{FR}}^{\mathcal{P}}} \mathcal{L}(p(\theta)), \tag{18}$$

in which we just use the classical definition of the gradient on a manifold.

This means that in the approximation of $\epsilon \to 0$ the step $p(\theta) \to p(\theta + \delta\theta)$ must move in the negative-$g_{\mathrm{FR}}^{\mathcal{P}}$-gradient direction of $\mathcal{L}$ at $p(\theta)$, as desired. $\qquad\square$

## D  Optimal Transport proofs

**Proposition 4** (extended version of Proposition 2). *For any two Borel probability measures $p_0, p_1 \in \mathcal{P}(\mathbb{S}_+)$, the following hold:*

1. *There exists a unique OT-plan $\pi$ between $p_0, p_1$.*

2. *For $t \in [0,1]$ let $e_t(x_0, x_1)$ be the constant-speed parameterisation of the unique geodesic of extremes $x_0$ and $x_1$, defining the map*

$$e_t : \mathbb{S}_+ \times \mathbb{S}_+ \to \mathbb{S}_+, \quad e_t(x_0, x_1) \coloneqq \exp_{x_0}(t \log_{x_0}(x_1)). \tag{19}$$

   *Then there exists a unique Wasserstein geodesic $(p_t)_{t \in [0,1]}$ connecting $p_0$ to $p_1$, and it is given by*

$$p_t \coloneqq (e_t)_{\#} \pi \in \mathcal{P}(\mathbb{S}_+), \quad t \in [0,1]. \tag{20}$$

3. *For every point $x_t$ in the support of $p_t$, there exists a unique pair $(x_0, x_1)$ in the support of the optimal transport plan $\pi$ such that $x_t = e_t(x_0, x_1)$. Furthermore, the assignment $x_t \mapsto (x_0, x_1)$ is continuous in $x_t$.*

4. *The probability path $(p_t)_{t \in [0,1]}$ has velocity field $u_t \coloneqq \log_{x_t}(x_1) - \log_{x_t}(x_0)$, which is uniquely determined over the support of $p_t$.*

5. *The above probability measure path and associated velocity fields $(p_t, u_t)_{t \in [0,1]}$ are minimisers of the following kinetic energy minimisation problem*

$$\min_{(\rho_t, v_t)_{t \in [0,1]}} \left\{ \int_0^1 \mathbb{E}_{\rho_t}[\|v_t\|^2]\mathrm{d}t : \partial_t \rho_t + \mathrm{div}(\rho_t v_t) = 0, \quad \rho_0 = p_0, \ \rho_1 = p_1 \right\}. \tag{21}$$

*Proof.* For point 1, we can use Villani [76, Thm. 10.28] (the simpler Villani [76, Thm. 10.41] also applies, with the minor modification that we work on a manifold with boundary). To verify its conditions, note that $\mathcal{M}^d \subset \mathbb{S}_+$ is a subset of a Riemannian manifold and has $(d-1)$-dimensional measure, and that cost $c(x, y) = d^2(x, y)$ is convex, thus it has unique superdifferential and $\nabla_x c(x, \cdot)$ is injective, as required.

For points 2 and 3, we note that by Villani [76, Cor. 7.22] (see also McCann [54]), in general Polish spaces displacement interpolants as given by Eq. 19 and Eq. 20, coincide with Wasserstein geodesics.

A simplified version of the proof of 4. is present in Santambrogio [63, Prop. 5.30]. For the general case, we can use Villani [76, Thm. 10.28], in particular eq. (10.20) therein. Note that for $c(x, y) = d^2(x, y)$, as indicated in Example 10.36 this equation corresponds to the equation of geodesics in the underlying manifold. Then we just note that $u_t$ is the velocity field of a constant speed geodesic.

Point 5 is a special case of Villani [76, Thm. 7.21], see also Granieri [33].

$\qquad\square$

## E  Relation to prior work on the simplex

### E.1  Dirichlet Flow matching

In this appendix, we discuss how flow matching can be done on the simplex using Dirichlet conditional probability paths. This recovers the simplex flows designed in Stark et al. [68], Campbell et al. [23].

The equivalent of a uniform density over $\Delta^d$ is given by a Dirichlet distribution with parameter vector $\alpha = \mathbf{1}$, *i.e.*, $p_1(x_1) = \mathrm{Dir}(x_1; \alpha = \mathbf{1})$. This is the starting point for defining a flow between our data

distribution, $p_0$, and the Dirichlet prior $p_1$. As proven in Stark et al. [68] we can reformulate Eq. 3 using a cross-entropy objective,

$$\mathcal{L}_{\text{ce}}(\theta) = \mathbb{E}_{t,q(z),p_t(x_t|z)} \|v_\theta(t, x_t) - u_t(x_t|z)\|_g^2 \tag{22}$$

$$= \mathbb{E}_{t,q(z),p_t(x_t|z)} \| \log \hat{p}_\theta(x_0|x_t) \|_g^2. \tag{23}$$

Here, we parameterise a *denoising classifier* which predicts a denoised sample $x_0$ from $x_t$, which is built using the conditioner $z$. Such a parameterisation naturally restricts the vector field to move tangentially to the simplex and also training is simplified as we do not need to explicitly construct the conditional vector field $u_t(x_t|z)$ during training. At inference, we can recover $v_\theta(t, x_t) = \sum_i^d u_t(x_t|x_0 = e_i)\hat{p}_\theta(x_0 = e_1|x_t)$ and follow the $v_\theta$ by integrating time to $t = 1$.

**Designing conditional paths**. There are two primary points of attack when designing a flow-matching model. We can either define an interpolant $\psi_t(x_t|z)$ with initial conditions $\psi_0 = x_0$, which we can differentiate to obtain $u_t$, *i.e.*, $\dot{\psi}(x_t|z) = u_t(x_t|z)$; or we can operate on the distributional level and specify conditionals $p_t(x_t|z)$ from which a suitable vector field can be recovered.

If we take the interpolant perspective, one can easily implement the linear interpolant [48, 73], which gives the following conditional vector field:

$$\psi_t(x_t|x_0, x_1) = tx_0 + (1-t)x_1 \tag{24}$$

$$u_t(x_t|x_0, x_1) = \frac{x_t - x_0}{t} = x_0 - x_1 \tag{25}$$

Unfortunately, in the case of flow matching on the simplex, the linear interpolant has undesirable properties in that the intermediate distribution induced by the flow must quickly reduce support over $\Delta^d$ by dropping vertices progressively for $t > 0$ [68].

Operating directly on the distribution level, we can define $p_t$ as themselves being Dirichlet distributions indexed by $t$ such that, at $t = 0$, we have a uniform mass over $\Delta^d$, and that, at $t = 1$, we reach a vertex. One choice of parameterisation that fulfills these desiderata is

$$p_t(x_t|x_0 = e_i) = \text{Dir}(x_t; \alpha = 1 + t' \cdot e_i), \tag{26}$$

where $t' = f(t)$ is a monotonic function of the original time variable $t$ such that $f(0) = 0$ and $\lim_{t \to 1^-} f(t) = \infty$. Clearly, $t' = 0$ recovers the uniform prior as $\alpha = \mathbf{1}^\top$, while $t' \to \infty$ increases the mass of $e_i$ while other vertices remain constant. Given the conditional in Eq. 26, one corresponding vector field that satisfies the continuity equation is

$$u_t(x_t|x_0 = e_i) = C(x_i, t)(x_t - e_i), \quad C([x_t]_i, t) = -\tilde{I}_{x_i}(t+1, d-1) \frac{\mathcal{B}(t+1, d-1)}{(1-x_i)^{d-1}x_i^t}, \tag{27}$$

where $\tilde{I}_x(a, b) = \frac{\partial}{\partial a} I_x(a, b)$ is the derivative of the regularised incomplete beta function [68, Appendix A.1] and $\dot{C} \propto 1/t$ as in regular linear flow matching.

### E.2 $e$-geodesics on the Assignment manifold

In this appendix, we survey other common geometries implied by the theory of $\alpha$-divergences on statistical manifolds, described in more detail in Amari [4, Ch. 4] or Ay et al. [11, Ch. 2], of which the case of $e$-connections was proposed in relation to flow-matching in Boll et al. [17].

In what has become a fundamental paper for the field of Information Geometry, Amari [3] unified several commonly used parameterisations of statistical manifolds, in the theory of so-called $\alpha$-connections. Without entering full details (which can be found in the mentioned references), on a statistical manifold, endowed with Fisher-Rao metric, one can introduce a 1-parameter family of affine connections, so-called $\alpha$-connections with $\alpha \in [-1, 1]$, where $\alpha = 0$ corresponds to Fisher-Rao Levi-Civita connection, and other notable values are the $m$-connection for $\alpha = -1$ and the $e$-connection for $\alpha = 1$. Furthermore, specific classes of $\alpha$-divergences – which for $\alpha = 0$ recover KL divergence – have been introduced as adapted to the corresponding $\alpha$-connections.

**Algorithm 1** FISHER-FLOW, training on $\mathbb{S}_+^d$.

---

1: **Input:** Source and target distributions, $p_1, p_0$, flow network $v_\theta$.
2: **while** Training **do**
3:     $t, x_0, x_1 \sim \mathcal{U}(0,1), p_0, p_1 = p_{\text{data}}$
4:     $\bar{\pi} \leftarrow \text{OT}_{\mathbb{S}_+^d}(x_0, x_1)$               $\triangleright$ *Since $x_1$ is one-hot encoded, it is on $\mathbb{S}_+^d$.*
5:     $x_0, x_1 \sim \bar{\pi}$
6:     $x_t \leftarrow \exp_{x_0}(t \log_{x_0}(x_1))$         $\triangleright$ *Geodesic interpolant between $r_0, r_1 \in \mathbb{S}_+^d$.*
7:     $u_t(x_t|x_0, x_1) \leftarrow \dot{x}_t$     $\triangleright$ *Calculated either explicitly or with a numerical approximation.*
8:     $\mathcal{L}_{\text{FISHER-FLOW}} \leftarrow \|v_\theta(t, x_t) - u_t(x_t|x_0, x_1)\|_{\mathbb{S}_+^d}^2$
9:     $\theta \leftarrow \text{Update}(\theta, \nabla_\theta \mathcal{L}_{\text{FISHER-FLOW}})$
10: **return** $v_\theta$

---

In general, a choice of differential geometric connection allows to define ad-hoc covariant derivatives, and corresponds to an explicit formula for associated geodesics (curves whose tangent vector has zero covariant derivative).

For the case of $\alpha$-connections on categorical probabilities $\mathcal{P}(\mathcal{A})$, explicit formulas can be given (see Ay et al. [11, Ch. 2]), recovering, for $m$-connections, interpretations as mixtures, with geodesics equal to straight lines in $\Delta^d$-parameterisation, and for $e$-connections geodesics can be interpreted as exponential mixtures, as elucidated in Ay et al. [11, Ch. 2] and illustrated in Boll et al. [17].

For the case of $e$-connections, concurrent work [17] has proposed to use the corresponding explicit parameterisation of geodesics in flow-matching, leaving as an open question the adaptation of Optimal Transport ideas to the framework.

## F   Implementation Details

### F.1   General Remarks

All of our code is implemented in Python, using `PyTorch`. For the implementation of the manifold functions (such as $\log$, $\exp$, geodesic distance, etc.), we have tried two different versions. The first one was a direct port of `Manifolds.JL` [10], originally written in Julia; the second one used the `geoopt` library [46] as a back-end. The latter performed noticeably better—the underlying reason being probably a better numerical stability of the provided functions.

As for the optimal transport part, it is essentially an adaptation of that of FoldFlow [18], which itself relies on the `POT` library [31].

### F.2   FISHER-FLOW Algorithm

We provide pseudo-code for training FISHER-FLOW Algorithm 1.

### F.3   Compute Resources

All experiments are run on a single Nvidia A10 or RTX A6000 GPUs.

### F.4   Experiments

#### F.4.1   Toy Experiment

We reproduce most hyper-parameters, except for the number of epochs trained for 500 instead of 540,000. Nonetheless, a *major* modification from the original setting is the size of the dataset. Indeed, in the original dataset code of Stark et al. [68][5], one can observe that the points are generated at each retrieval, and the defined length of the dataset is of $10^9$, thus amounting to $540{,}000 \cdot 10^9$ training

---

[5]https://github.com/HannesStark/dirichlet-flow-matching/blob/main/utils/dataset.py#L53, retrieved on October 30, 2024.

points by the end of the training process. This results in an unrealistic learning setup. To slightly toughen the experiment, we limit the training set size to 100,000 points.

Note that the model with which we train our method is a much simpler architecture than that of DIRICHLET FM (which was the one used in Stark et al. [68]), ours consisting exclusively of (residual) MLPs. For lower dimensions, it has less parameters, and slightly more in higher dimensions. The other baselines were run with our MLP too.

Table 5: Fréchet Biological Distance (FBD) and perplexities (PPL) values for different methods for enhancer DNA generation. Lower FBD and PPL are better. Values are an average and standard error over 5 different runs.

| Method | Melanoma FBD ($\downarrow$) | Melanoma PPL ($\downarrow$) | Fly Brain FBD ($\downarrow$) | Fly Brain PPL ($\downarrow$) |
|---|---|---|---|---|
| Random Sequence | $619.0 \pm 0.8$ | $895.88$ | $832.4 \pm 0.3$ | $895.88$ |
| Language Model | $35.4 \pm 0.5$ | $2.22 \pm 0.09$ | $25.7 \pm 1.0$ | $2.19 \pm 0.10$ |
| DIRICHLET FM | $\mathbf{7.3 \pm 1.2}$ | $2.25 \pm 0.01$ | $6.8 \pm 1.8$ | $2.25 \pm 0.02$ |
| FISHER-FLOW (ours) | $27.5 \pm 2.6$ | $\mathbf{1.4 \pm 0.1}$ | $\mathbf{3.8 \pm 0.3}$ | $\mathbf{1.4 \pm 0.66}$ |

### F.4.2  Promoter DNA

We train our generative models for 200,000 steps with a batch size of 256. We cache the best checkpoint over the course of training according to the validation $MSE$ between the true promoter signal and the signal from the Sei model conditioned on the generated promoter DNA sequences. We use the same train/val/test splits as Stark et al. [68] of size 88,470/3,933/7,497.

The generative model used for FISHER-FLOW and DFM Stark et al. [68] is a 20 layer 1-d CNN with an initial embedding for the DNA. Each block consists of a LayerNorm [12] followed by a convolutional layer with kernel size 9 and ReLU activation and a residual connection. As we stack the layers we increase the dilation and padding of the convolutional allowing the receptive field to grow [57]. In general, we use the AdamW optimiser [49].

Our Language Model implementation is identical to Stark et al. [68] and we use the pre-trained checkpoint provided by the authors and evaluated on the test set.

### F.4.3  Enhancer DNA

We consider two DNA enhancer datasets, the fly brain enhancer dataset with 81 classes [45], the classes are different cell types, and the melanoma enhancer dataset with 47 classes [7]. Both datasets are comprised of DNA sequences of length 500. We use the same train/val/test splits as Stark et al. [68] of size 70,892/8,966/9,012 for the human melanoma and 83,726/10,505/10,434 for the fly brain enhancer DNA dataset.

The generative model for our experiments for FISHER-FLOW and DFM [68] is the same as used for our promoter DNA experiments. Specifically, we use a 20 layer 1-d CNN with an initial embedding for the DNA. Each block consists of a LayerNorm [12] followed by a convolutional layer with kernel size 9 and ReLU activation followed by a residual connection. As we stack the layers we increase the dilation and padding of the convolutional allowing the receptive field to grow [57].

We train for a total of 450,000 steps with a batch size of 256 where we cache the best checkpoint according to the validation FBD. The test set results in Table 2 are using the best checkpoint according to the validation FBD.

To calculate the FBD we compare embeddings from FISHER-FLOW with a shallow 5 layer classifier embeddings originally trained to classify the cell type given the enhancer DNA sequences. Our Language Model implementation is identical to Stark et al. [68] and we use the pre-trained checkpoint provided by the authors and evaluated on the test set.

### F.5  Additional Metrics

For complete transparency, we also report the Fréchet Biological Distance (FBD) for the DNA enhancer generation experiments as initially reported in Stark et al. [68].

The FBD computes Wasserstein distance between Gaussians of embeddings generated and training samples under a pre-trained classifier trained to predict cell-type of enhancer sequences. Versus those embeddings from the generative model under consideration. So crucially there is a dependence on classifier features.

On FlyBrain we find that FISHER-FLOW also improves over DFM in FBD being roughly $\approx 2\times$ better while DFM is better on Melanoma. However, we caveat both FBD results by noting the trained classifiers provided in DFM [68] obtain a test set accuracy of $11.5\%$ and $11.2\%$ on the Melanoma dataset and FlyBrain dataset respectively. Moreover, switching out the pre-trained classifier for another trained from scratch caused large variations in FBD metrics. As a result, the low-accuracy classifiers do not provide reliable representation spaces needed to compute FBD metrics. Consequently, FBD in this setting is a noisy metric that is loosely correlated with model performance, so we opt to report perplexities in Table 2.

### F.6 *De Novo* Molecule Generation

Following the setup of Dunn and Koes [29], we report the following metrics for our model on *de novo* molecule generation over the QM9 and GeomDrugs datasets: percentage of stable atoms, percentage of stable molecules, percentage of valid molecules. Note that the following inference scheme is used, when training a model $\hat{x}_1$ on endpoint prediction:

$$x_{t+1} = \exp_{x_t} \left( \alpha'(t) \frac{\Delta}{1 - \alpha(t)} \log_{x_t}(\hat{x}_1) \right), \tag{28}$$

where $\Delta = 1/N$, and $N > 0$ is the number of integration steps.

