# OpenReview forum: "Fisher Flow Matching for Generative Modeling over Discrete Data"
_NeurIPS.cc/2024/Conference — NeurIPS 2024 poster_

### Official Review · Reviewer_YHTA · 2024-07-12

**Soundness:** 3
**Presentation:** 4
**Contribution:** 2
**Rating:** 7
**Confidence:** 4

**Summary:**

The paper aims to build a Flow Matching [1] based generative model for discrete data. The approach models the discrete data as a categorical distribution that resides on the simplex, thereby translating the problem into continuous flows. Equipped with the Fisher-Rao metric, the simplex is identified as a Riemannian manifold, and the task is reduced to Riemannian Flow Matching [2]. To further simplify the geometry of the space, the sphere map is used, which identifies the simplex with the positive orthant of a hypersphere. Finally, the conditional path between every pair $(x_0, x_1)$ is defined via the geodesic interpolant induced by the FIsher-Rao metric. For marginalization, two joint distributions are considered: 1. $x_0$ and $x_1$ are independent, i.e., $\pi(x_0,x_1) = p_0(x_0)p_1(x_1)$ as in [2]. 2. $\pi(x_0,x_1)$ is the optimal transport (OT) plan between $p_0$ and $p_1$, approximated with minibatch [3].

**Strengths:**

1. The paper is well-written, and the method is rigorously presented.

2. The author successfully frames the proposed method for generative models of discrete data in terms of well-established existing methods, which eases the understanding of the approach and potentially allows for a relatively low cost of implementation.

3. Aside from the computational cost of the OT plan, the method does not introduce additional computational costs compared to previous works.

**Weaknesses:**

1. The use of Riemannian Optimal Transport with a mini batch is of low novelty. This has also been done in [3] and on euclidean space in  [6,7].

2. Two key contributions of the paper seems to be the change in geometry, that is the introduction of the Fisher-Rao metric and the sphere map. However the paper lacks a clear experimental  demonstration of the portion these two components contribute to results. Specifically:

    2.1. Regarding Figure 2, allegedly,  the author claim that Fisher-Flow (FF)-OT on $\mathbb{S}_+^2$ (e) performs best, however it is not clear, one may even say that FF-OT on the $\Delta^2$ (c) perform better. Furthermore, even without FF-OT it is not clear that using the sphere map helps ( (b) vs (d)).

    2.2. In Figure 3.a the author compares FF simplex, FF-OT simplex, FF sphere, and FF-OT sphere, that shows that FF-OT sphere preforms the best. It is also apparent that with increasing dimension size the advantage of the use of the OT plan decreases. And Figure 3.b shows that FF-OT sphere out performs [4]. However, comparing Figure 3.a and 3.b it seems that the other three, i.e., FF simplex, FF-OT simplex, FF sphere are actually outperformed by [4] which uses flows on the simplex but no OT plan. To conclude, this experiment suggests that the apparent advantage of FF-OT sphere on [4] may exist only for relatively low dimensions.

    2.3. The other two experiments only show FF-OT sphere.

    2.4. No ablation that focus on the use of Fisher-Rao metric compared to other potential metrics (though the author does provide a theoretical justification).

**Questions:**

1. Can the author expand on the difference of his use in Riemannian Optimal Transport with a mini batch compared to previous works?
2. The author claims that [5] is restricted in the choice of possible source distributions. Can the author please elaborate on that?
3. Could the author possibly provide more experimental evidence to demonstrate that the proposed change in geometry indeed contributes to a considerable improvement and that the gain in results is not mainly from the OT plan?

**Limitations:**

It is suspected that the proposed method might not be as effective in large dimensions (i.e., large vocabulary size ) compared to other methods.

[1] Lipman, Y., Chen, R. T. Q., Ben-Hamu, H., Nickel, M., & Le, M. (2023). Flow Matching for Generative Modeling. arXiv preprint arXiv:2210.02747.

[2] Chen, R. T. Q., & Lipman, Y. (2023). Flow Matching on General Geometries. arXiv preprint arXiv:2302.03660.

[3] Bose, A. J., Akhound-Sadegh, T., Huguet, G., Fatras, K., Rector-Brooks, J., Liu, C.-H., Nica, A. C., Korablyov, M., Bronstein, M., & Tong, A. (2024). SE(3)-Stochastic Flow Matching for Protein Backbone Generation. arXiv preprint arXiv:2310.02391.

[4] Stark, Hannes, et al. "Dirichlet flow matching with applications to dna sequence design." arXiv preprint arXiv:2402.05841 (2024).

[5] Campbell, Andrew, et al. "Generative flows on discrete state-spaces: Enabling multimodal flows with applications to protein co-design." arXiv preprint arXiv:2402.04997 (2024).

[6] Tong, Alexander, et al. "Improving and generalizing flow-based generative models with minibatch optimal transport." arXiv preprint arXiv:2302.00482 (2023).

[7] Pooladian, Aram-Alexandre, et al. "Multisample flow matching: Straightening flows with minibatch couplings." arXiv preprint arXiv:2304.14772 (2023).

---

> ### Author Rebuttal · Authors · 2024-08-07
>
> We thank the reviewer for their time and detailed feedback. We focus here on addressing the key clarification points raised in this review while the global response contains new experiments on molecules and language modelling with additional baselines.
> ## Riemannian OT is of low-novelty
> We acknowledge the reviewer’s concern that OT for Riemannian manifolds has previously been considered in [1]. Unfortunately, the general theory of optimal transport on manifolds Villani 2003 [2] (Theorem 2.47) does not give a prescription for a fast computation implementation of OT on manifolds—i.e., analytic expressions for the geodesic cost $c(x_0, x_1)$.
> While [1] found a closed form solution to this cost for $SE(3)$, it cannot be generally extended to all Riemannian manifolds, since, for arbitrary manifolds, one must measure the geodesic by simulating the expensive Euler-Lagrange equations, which are 2nd order PDEs.
>
> Our work with Fisher-Flows pushes the advantage of Riemannian OT within generative modelling by showing that:
> 1.) The probability simplex and the sphere both have closed form expressions for the OT cost $c(x_0, x_1)$.
> 2.) We demonstrate that OT via Fisher-Flows can be used for discrete data. We highlight that this is the first use of OT for generative modelling over discrete data.
> 3.) As proved in Appendix D, OT over the probability simplex elicits new benefits, such as the fact that $p_t$ becomes the Wasserstein geodesic between $p_0$ and $p_1$, and the associated velocity field $v_t$ minimises the kinetic energy. These two points were not rigorously proven in [1].
> ## Clearer experimental demonstration over the benefits of the Fisher-Rao metric and Sphere map
> ### OT in Fig 2
> With regards to Fig. 2, we find that both c) and e) to be of comparable visual quality, and, more importantly, of a higher one than that of their non-OT counterparts. As this is a small toy synthetic density estimation task, the benefits of the Fisher-Rao metric and the added numerical benefits of the sphere map are less pronounced.
> ### Comparison of FR on Fig 3
> Since the sphere is more numerically stable and, for practical problems—where the number of dimensions can greatly increase—we expect Fisher-Flows on the sphere to perform the best—regardless of OT. We see this in Fig 1 of the 1pg rebuttal PDF, Dirichlet Flow Matching is often worse than FF-no OT on the sphere as we increase the number of categories. With regards to OT, our experimental findings are in line with previous use of OT in generative models, where lowering variance in training and shorter paths during inference (fewer ODE integration steps [3,4]) are observed.
> ### The other two experiments only show FF-OT sphere.
> Our new experiments on molecule generation and language modelling (see 1pg rebuttal PDF) used Fisher-Flows on the sphere and without OT.
> In our language modelling results, we found that Fisher-Flow slightly outperforms discrete diffusion approaches (e.g., D3PM, SEDD) while marginally outperforming concurrent SOTA work on Masked Discrete Diffusion (MDLM [5]) released after the submission deadline. This suggests that geometry plays a key role in improvement as we compare against conventional discrete diffusion methods on higher dimensional problem setting ~800k categories (text).
> ### No ablation on other potential metrics.
> Thank you for this suggestion! We have included new ablations in our 1 pg rebuttal PDF that use the Euclidean metric (Linear Flow Matching) on the simplex for both the synthetic data and DNA Promoter and Enhancer experiments. We observe that the Fisher-Rao metric is still on par and better than Dirichlet FM, in terms of empirical performance. Finally, considering other metrics beyond the Fisher-Rao and Euclidean is an interesting thought; however, it is unclear whether it is possible to easily obtain closed-form geodesics, which are needed for simulation-free training as mandated in the flow matching framework.
> ## Q1. Use of Riemannian OT vs. Prior Work
> Please see our detailed response under the heading “Riemannian OT is of low-novelty” for a deeper discussion.
> ## Q2. Flexibility of source distribution in Campbell et. al 2024?
> We note that Campbell et. al 2024 explicitly state that the “conditional flows we use in this paper linearly interpolate towards $x_1$ from a uniform prior or an artificially introduced mask state, M.” They do not make any theoretical claims about the generality of their starting source distribution unlike we do for Fisher-Flows. While it is potentially possible to accommodate for a more flexible source distribution in their CTMC framework, it is not immediately clear how to do this easily. We will refine our discussion over Campbell et. al 2024 in our paper to highlight this aspect.
> ## Q3. Ablation on geometry vs. OT?
> Please see our previous response on Fisher-Flows without OT on language modelling.
> ## Closing comment
> We hope that our responses were sufficient in clarifying all the great questions asked by the reviewer. We thank the reviewer again for their time, and we politely encourage the reviewer to consider updating their score if they deem that our responses in this rebuttal along with the new experiments merit it.
> ## References
> [1] Bose et. al (2024). SE(3)-Stochastic Flow Matching for Protein Backbone Generation. arXiv preprint arXiv:2310.02391.
>
> [2] Villani. Optimal Transport: Old and New. Grundlehren der mathematischen Wissenschaften. Springer Berlin Heidelberg, 2008. ISBN 9783540710509.
>
> [3] Tong et. al. Improving and generalizing flow-based generative models with minibatch optimal transport. arXiv preprint 2302.00482, 2023.
>
> [4] Klein et. al . "Equivariant flow matching." Advances in Neural Information Processing Systems 36 (2024).
>
> [5] Sahoo et al. "Simple and Effective Masked Diffusion Language Models." arXiv preprint arXiv:2406.07524 (2024).

---

> ### Comment · Reviewer_YHTA · 2024-08-11
> **Reviewer response**
>
> I want to thank the authors for their efforts in adding more experiments.
>
> First regarding the Riemannian OT, thank you for emphasizing the contributions upon previous works. I am now convinced that more credit should be attributed to the authors than I initially acknowledged. Second regarding the significance of change in geometry from simplex to sphere, Table 1 in pdf actually shows no advantage to the sphere and to my understanding Table 2 compares only to discrete diffusion methods (where I would expect comparison like Table 1 to support such a claim.) However, I do find the work in this paper interesting and well presented, I will increase my score.

---

### Official Review · Reviewer_cNL3 · 2024-07-15

**Soundness:** 3
**Presentation:** 3
**Contribution:** 3
**Rating:** 6
**Confidence:** 4

**Summary:**

This paper proposes a framework that enables flow matching over a d-dimensional simplex, by instantiating a riemannian flow matching algorithm using the Fisher-Rao metric. Some motivation in connecting the Fisher-Rao metric to natural gradient descent and Riemannian optimal transport is used to justify the choice of Fisher-Rao metric. Experimental validation is carried out on DNA sequences.

**Strengths:**

The proposed method is quite a natural application of riemmanian flow matching to distributions over simplices.

**Weaknesses:**

- I felt the writing is a bit too mathematically dense, and the justifications for the Fisher-Rao metric feel a bit circular (see below for questions on Propositions 1 & 2).
- Additional ablation experiments on non-toy data would be good to have to verify some of the claims in the paper.
- It's unclear to me (due to lack of expertise in the application area) to judge how well this approach performs.

**Questions:**

### Regarding the mathematical justifications

I find that Sections 3.3 and 3.4 are circular in their reasoning.

In Section 3.3, it is stated "the choice of Fisher-Rao metric [is] the optimal one on the probability simplex". However, here "optimality" is defined in terms of the expected KL, W_KL. This is an arbitrary choice of optimality. There is obviously a correspondence between a distance function and a Riemannian metric, and any metric can be stated as the optimal one for some distance function. So:
- Why is considering the W_KL ball in Eq 8 useful? Are there computational reasons to use this? Are there theoretical justifications over other distance functions?

In Section 3.4, a similar problem arises. Here it is stated that the flows (1) lead to shorter global paths and (3) have lower kinetic energy. However, this is circular because the distance of a path (and consequently, the kinetic energy) is an implication of the chosen Riemannian metric g. Here the correspondence between distance function <==> Riemannian metric shows up again.
- Why do we care about the lengths of paths defined by the Fisher-Rao metric?
- Does the choice of Fisher-Rao metric give velocity fields that are faster to simulate?

I'm not completely sure what the point of Proposition 2 is. Equation 10 looks to me like a definition of p_t given a coupling pi. That is, given that we've already solved the optimal transport coupling, we can then take any probability path p_t that transports between the marginals of pi, so Proposition 2 is just stating a definition of a particular choice of p_t.
- Here do you mean to imply something about this choice of p_t? E.g., that it allows expressing W_2 in terms of p_t through a dynamical formulation, where the kinetic energy (as defined by the Fisher-Rao metric) shows up?

### Regarding empirical validation

The paper makes the following claims:
(1) flexibility of source distribution
(2) the sphere map has better numerical stability properties
(3) the choice of Fisher-Rao metric enables continuous reparameterisation and OT

The experiments do not yet completely justify these points, from what I can tell.

(1) There is no experiment that uses a flexible source distribution.
- Is there a case for using something that isn't a Dirichlet distribution which can be handled by Dirichlet FM? Or at least, a justification over the uniform noise distribution?

(2) It's unclear why the sphere map helps. Concretely, there is a division by p in the computation of the inner product (Eq 5). However, this inner product is unneeded for the geodesic computation, since we can easily solve the geodesic on the sphere and then map back to the simplex. So this inner product seems to show up only in Eq 7, the training objective. However, given that this metric only depends on x_t, and that the model conditions on x_t, the optimal solution of CFM (Eq 7) is unaffected. So one can easily use any other metric for CFM without affecting the optimal solution.
- Have the authors tried just using another metric for CFM and training on the simplex representation?

(3) Similarly, I think a good ablation to do is just to use the Euclidean metric here. One can easily use the interpolants defined by the Fisher-Rao metric instead of linear, but otherwise just use regular Euclidean flow matching. Otherwise, related to the first concern, it is unclear why the connections to kinetic energy (as in Riemannian optimal transport) is important.

### Clarify about experiment setup

As someone not familiar with DNA sequences, the experiment section does not provide enough information for me to gauge the usefulness of these experiments. For instance, it should at least be stated what d is (I imagine d=4?). My subjective opinion is that many methods could work reasonable well with such a small d. It would be even better if the authors could comment on the real world implications of their improved metrics (how does PPL affect a scientist who would use this generative model?)

**Limitations:**

The paper discusses a bit the limitations regarding higher d, but does not provide details on why the current approach cannot be applied. I am also curious why the authors did not consider relatively toy datasets (such as variants of MNIST) that would be more understandable / appealing for the general machine learning community. For instance, it is unclear to me if non-DNA experiments would be too hard, or even too easy, compared to the chosen experiments.

---

> ### Author Rebuttal · Authors · 2024-08-07
>
> We thank the reviewer for their time and effort assessing our paper. We now address the key points in the review, while additional experiments are included in the global response and rebuttal PDF.
> ## Circular justifications for the Fisher Rao
> We understand that certain aspects of the theory could be explained more clearly. We now explain our reasoning step by step. First, note that a distance function on a Riemannian manifold is fully determined by the choice of metric, and that choosing a different distance function amounts to choosing a different metric.
>
> We require the following key desiderata from a metric to employ flow matching:
> - Analytic expressions of manifold operations like the exponential and logarithmic map.
> - Easy parameterisation of the tangent space in order to define vector fields.
> - Interpretability and naturality, i.e. connection to established statistical and physical theory.
> - Numerically stability of the metric over the entire manifold.
>
> The use of the FR metric satisfies both points 1 and 2 above and more and point 4 is achieved via the sphere map. In addition, we argue that the pervasiveness of the use of the forward KL divergence in generative models such as diffusion, flows, and VAE, testifies to its naturalness. As to point 3, the KL divergence is natural due to its strong link to fundamental concepts from Information Theory, being interpretable in terms of entropy. In this sense, KL is perhaps the simplest natural divergence over categorical probabilities.
>
> We now explain why our choice does not lead to circular reasoning using Prop 1.
> Note that Prop 1 starts off with hypotheses over a generic metric “g” in the $W_{2,g}$ distance from the hypothesis. The thesis of the proposition shows that natural gradient with respect to KL-divergence-based $W_{KL}$ distance produces naturally the $W_{2, g_{FR}}$ gradient flow, with now “$g_{FR}$” being FR metric, which was not assumed in the hypotheses. Thus this is not a circular statement, as it deduces FR as the limit of KL-divergence, at the level of probability spaces.
>
> >Are there computational reasons to use this?
>
> The FR metric leads to theoretical benefits as it is also the Wasserstein geodesic over probability paths on the simplex. The Euclidean metric, as explored in Dirichlet Flow Matching, does not enjoy such benefits and arguably is one reason that is empirically less performant than Fisher-Flows.
>
> >In Section 3.4, a similar problem arises…
>
> We believe there may be a slight misunderstanding as the logic of our setup is to first pick a metric and then show OT is computationally feasible. Indeed other metrics are possible and will lead to different OT maps; but, crucially, they may not be easy to compute for generative models. Fortunately, for the FR metric the distance is simply the distance on a sphere, which is **easy to compute** allowing us to efficiently compute the OT cost $c(x_0, x_1)$.
>
> >Why do we care about the lengths of paths defined by the Fisher-Rao metric? … velocity fields faster to simulate?
>
> Shorter paths lead to lower variance training: on the sphere and the simplex. The paths do not cross at intermediate points $x_t$, which would have added to noisier training—in line with the literature see [1,2]. We note OT often leads to vector fields that are faster to simulate using numerical solvers as they incur smaller per step error, as studied for $SE(3)$ in [3].
>
> >Here do you mean to imply something about this choice of p_t? …
> Prop. 2 shows that, by using FR geodesics as our flow, we automatically use the geodesic path between $p_0$ and $p_1$ with respect to the natural Wasserstein metric over probability measures. A consequence of this is that the links to kinetic energy and dynamical formulations apply (see Prop. 4).
> ## Empirical validation
> >There is no experiment that uses a flexible source distribution
> We invite the reviewer to kindly read our global response which includes new experiments over text that uses a new source distribution which is a convex combination of a specialised mask token and the uniform distribution.
>
> >It's unclear why the sphere map helps ….
>
> The metric appears when we compute $x_t$ due to exp and log map on the simplex (see Appendix B). Looking at equation 11, for the exp map, we notice we need to compute the norm which is induced by the Riemannian metric so $<v^2_p, v^2_p>_{FR}$. Similarly, the log map on the simplex in equation 12 explicitly requires the inner product. Exploiting the sphere map, we obtain $x_t$ and avoid numerical instability near the boundary of the manifold. Other metrics beyond the FR are indeed possible on the simplex but it may not have analytic formulae for $x_t$.
>
> >Similarly, I think a good ablation to do is just to use the Euclidean metric here …
>
> Please see our global response PDF where we have included the Euclidean metric in Linear Flow Matching on the toy datasets as well as our DNA experiments.
> ## Clarifying experimental setup
> In addition to DNA sequences we have now also included more standard generative modelling experiments on molecules and larger scale language modelling that demonstrate the scalability of Fisher-Flows on a host of different domains.
>
> We thank the reviewer again for their valuable feedback. We hope that our rebuttal addresses their questions and concerns, and we kindly ask the reviewer to consider fresher evaluation of our paper if the reviewer is satisfied with our responses. We are also more than happy to answer any further questions that arise.
> ## References
> [1] Bose et. al (2024). SE(3)-Stochastic Flow Matching for Protein Backbone Generation. arXiv preprint arXiv:2310.02391.
>
> [2] Tong et. al. Improving and generalizing flow-based generative models with minibatch optimal transport. arXiv preprint 2302.00482, 2023.
>
> [3] Klein et. al  "Equivariant flow matching." Advances in Neural Information Processing Systems 36 (2024).

---

> > ### Author Response · Authors · 2024-08-12
> > **Kindly awaiting more feedback**
> >
> > We thank you again for your time and feedback that allowed us to strengthen the paper with new experiments. As the end of the rebuttal period is fast approaching we were wondering if our answers in the rebuttal were sufficient enough to address the important concerns raised regarding 1.) the justification of using the Fisher-Rao metric and 2.) Empirical validation. We note that for point 2.) we included a host of new experiments, including a large-scale experiment on text on the LM1B dataset where we achieved SOTA among discrete diffusion and flow matching models.
> >
> > We would be happy to engage in any further discussion that the reviewer finds pertinent, please let us know! Finally, we are very appreciative of your time and effort in this rebuttal period and hope our answers are detailed enough to convince you to potentially upgrade your score if you believe it's merited.

---

### Official Review · Reviewer_11qA · 2024-07-16

**Soundness:** 3
**Presentation:** 2
**Contribution:** 3
**Rating:** 5
**Confidence:** 4

**Summary:**

The paper proposed a novel flow-based generative model called *Fisher-Flow* for discrete data. The model uses the Fisher metric to deduce a Riemmanian geometric structure of the statistical manifold. The authors also demonstrated the connections to natural gradient descent and optimal transport. Experiments on DNA datasets demonstrated better performance.

**Strengths:**

1. The geometric perspective of the manifold of categorical distributions, to the best of my knowledge, is a novel extension of flow matching models for discrete data. The use of the Fisher-Rao metric can induce a Riemannian structure for continuous parameterization.

2. The authors proposed a diffeomorphism to the unit sphere to avoid numerical instability during training and demonstrated better performance in experiments.


3. The authors provided connections between the proposed model and natural gradient descent and Riemannian optimal transport with mathematical proof. The proposed model can potentially share such theoretical benefits. Ablation studies also supported such claims.

**Weaknesses:**

1. Although being claimed as a general discrete generation model, **the proposed model was only tested on the specific task of DNA generation without further justification of its performance on more complex data such as image and text generation**. The two DNA datasets used in the paper have a small cardinality of 4 categories. Existing diffusion- or flow-based discrete generation models (e.g. BFN [1], D3PM [2], SEDD [3]) demonstrated good results on image generation (discretized pixel values or VQVAE tokens) and/or text generation (tokens) tasks which have higher cardinalities to show their scalability. It remains unclear whether the proposed model can be adapted for more general real-world tasks with a larger number of categories.

2. The choice of baselines is **too limited to be convincing enough** to demonstrate the proposed model's superior performance.
   1. For the density estimation task, **no baseline was compared** (results were more of an ablation study). For the toy data in higher dimensions, **only Dirichlet FM was compared**. For these two tasks on simplex, models like multinomial flow [4] should be tested to provide a comparison.
   2. For the enhancer DNA design task, **only Dirichlet FM was compared as a flow-/diffusion-based baseline**, despite that many other discrete diffusion models have been proposed, e.g., BFN [1], D3PM [2], SEDD [3]. Some of them were included in the related work section but none was tested as baselines in this task.
   3. Moreover, one simple but effective and commonly used baseline is missing across the board, which is linear flow matching on simplex (which was used in DirichletFM [5] and other multimodal flow matching papers as well).

3. Another critical concern lies in the **inconsistency of the reported performance of the same baseline models in this paper and the original paper**. Numbers for the Dirichlet FM reported in this paper for the promoter design task are **significantly worse** than those in the original paper. The original paper reported an MSE of 0.0269 for their best model (Table 1 in [5]) which is far better than the number indicated in this paper (0.034, Table 1 in this paper). Moreover, the linear flow matching model that achieved 0.0281 in Table 1 of [5] (outperforming the MSE of FisherFlow) was dropped in this paper. Similar large discrepancies in the baseline model's performance can be also noted in the FBD scores for enhancer design. Comparing Table 2 in [5] and Table 3 in this paper, the performance for the baseline Dirichlet FM was also significantly worse. As the numbers for other baselines were directly copied without change, it is unclear why the authors reported DirichletFM's performance differently. **Using the original MSEs or scores, the results indicated that the proposed model was not as good as Dirichlet FM or even Linear FM**.

4. In terms of evaluation metrics, **it is unclear whether perplexity is a valid metric** to compare flow/diffusion based models against language models. Perplexity was initially proposed for autoregressive language models, in which it was calculated directly over conditional probabilities of discrete tokens:
$$
PPL(X)=\exp\left(-\frac{1}{N}\sum_{k=1}^N\log p_\theta(x_k|x_{<k})\right)
$$
However, in the continuous flow/diffusion setting, such probabilities may **not be well-defined** as they are neither autoregressive nor defined over discrete space. Therefore, their PPL cannot be calculated in this way. If the authors instead tried to calculate the joint distribution $\log p_\theta(x_{1:N})$, it remains unclear how to leverage the unconditional flow model to calculate this log-likelihood for arbitrary given input sequence $x_{1:N}$, as the flow is defined over the continuous simplex space with infinite possible initial probability samples $p_0$. The authors did not explain how they were able to calculate the PPL in the paper, and it is unclear whether the derived PPL is comparable to the ones computed for autoregressive models.


[1] Graves, Alex, et al. "Bayesian flow networks." *arXiv preprint arXiv:2308.07037* (2023).

[2] Austin, Jacob, et al. "Structured denoising diffusion models in discrete state-spaces." *Advances in Neural Information Processing Systems* 34 (2021): 17981-17993.

[3] Lou, Aaron, Chenlin Meng, and Stefano Ermon. "Discrete diffusion language modeling by estimating the ratios of the data distribution." arXiv preprint arXiv:2310.16834 (2023).

[4] Hoogeboom, Emiel, et al. "Argmax flows and multinomial diffusion: Learning categorical distributions." *Advances in Neural Information Processing Systems* 34 (2021): 12454-12465.

[5] Stark, Hannes, et al. "Dirichlet flow matching with applications to dna sequence design." *arXiv preprint arXiv:2402.05841* (2024).
directly

**Questions:**

1. Previous existing diffusion- or flow-based discrete generative models have achieved considerable success in image and text domains (see Weakness 1). As a general discrete generation model, will the proposed model also achieve comparable performance on these image or text generation tasks? Will it have scalability issues regarding datasets with a large number of categories? More experiments on other generative domains will be needed to demonstrate the model's effectiveness and scalability.
2. Can you provide more baselines in the visualization of the density estimation toy dataset and some quantitative evaluation metrics for this task to better demonstrate the generation quality? With the wide range of existing discrete diffusion or flow based models, more should also be compared as baselines for both toy examples and the enhancer generation task. Specifically, will the proposed model outperform simple linear flow matching? See Weakness 2 for some existing diffusion or flow baselines.
3. Why the original numbers from the Dirichlet FM paper were not used? If you reran the experiments in the Dirichlet FM paper, can you provide justifications for why you only reran the ones for Dirichlet FM but not for other baselines like linear flow matching? It is also unclear why the results of linear flow matching in the original Dirichlet FM paper were omitted -- it has better performance than FisherFlow according to the Dirichlet FM paper. The reported numbers in this paper were **significantly worse** than those in the original paper. If original numbers were used, the proposed model cannot outperform the Dirichlet FM model. See Weakness 3 for details.
4. How was perplexity calculated for flow-based models? Is it calculated in an autoregressive fashion as the language models? (See Weakness 4) Flow-based model is not an autoregressive model. During generation, all tokens are simultaneously denoised into meaningful sequences. If you instead modeled the joint distribution $\log p_\theta(x_{1:N})$, how was this marginal log-likelihood calculated for arbitrary input sequence $x_{1:N}$? How can you deal with the randomness in the initial samples $p_0$ if a deterministic cross-entropy loss needs to be computed? Furthermore, how was perplexity calculated for random sequences? The log probability $\log p(x)$ for one-hot random sequences should either be 0 or infinite.
5. OT seems to lead to better performance. Will it significantly add to the computation time during training?
6. There is a notation inconsistency in the paper about whether to refer to the Dirichlet flow matching model as *Dirichlet FM* (e.g., Table 1 & 2) or *DFM* (main text).

**Limitations:**

Limitations were mentioned in the conclusion section, and the potential negative societal impact was adequately addressed.

---

> ### Author Rebuttal · Authors · 2024-08-07
>
> We would like to thank the reviewer for the time and effort they spent on reviewing our work. We are appreciative of the fact that the reviewer found our geometric perspective a “novel extension” to flow matching and that this leads to numerically stable training which is supported by better empirical performance. We now address the concerns raised by the reviewer.
> ## Experiments beyond DNA
> We acknowledge the reviewer's valid concern. In this rebuttal we included two more standard experimental settings for discrete data in molecular generation (QM9) and language modelling over text (LM1B). We kindly invite the reviewer to read our global response, and the 1pg rebuttal PDF, which gives more detail on these experiments. In summary, we found Fisher-Flows to be scalable to over 800k categories in text and achieve the best upper bound to text PPL amongst discrete diffusion models. For QM9 our results outperform flow matching and are competitive, almost saturating the task, with molecular diffusion models. We hope that our new experiments sufficiently demonstrates the scalability of Fisher-Flows to much higher dimensions and across a greater breadth of domains.
> ## Choice of baselines
> As suggested by the reviewer we’ve now included multiple baselines as part of our 1pg rebuttal PDF. Specifically, we added Linear Flow Matching (Euclidean metric) and Multinomial Flows to the synthetic task, where we found the latter to be numerically unstable as we increased the number of categories. We also added Linear Flow Matching to our DNA Promoter and Enhancer experiments where we found Fisher-Flows to be on par. For our new language modelling experiment on LM1B we compared against discrete diffusion models such as SEDD as well D3PM, as well as SOTA concurrent work in masked discrete diffusion in MDLM [1]. We found that Fisher-Flows still outperforms previous methods like D3PM and SEDD and is marginally better than MDLM.
>
> ## Reporting of Dirichlet Flow Matching results
>
> >“Why the original numbers from the Dirichlet FM paper were not used?
>
> Using the reference code implementation of DFM we noticed severe methodological issues that made it clear that the original numbers in their paper were an inaccurate reflection of true performance. We found the following flaws in their experimental setup:
> We found for the DNA datasets the provided command used `--subset_train_as_val` argument which sets the validation set as the training set. This meant that their reported metrics were measured on the training set.
> We found their released DNA classifier to have only $11%$ test accuracy which makes it a poor choice as a representation space to compute FBD.
> Their synthetic data experiment did not standardise a training and validation split and meant that their training set size was a function of run time—effectively infinite. Operationally, DFM was trained for $10^{10}$ points per epoch which is a tremendous amount for a synthetic task. We standardised the experiment to train all models with a more modest 10^6 points.
>
> After fixing these issues we reran DFM and found that it performed worse than what was initially reported. We are appreciative of the DFM authors for releasing their code that enabled us to find these experimental flaws, some which we outlined in Appendix F.4.2 and F.4.3 but will discuss further in our updated manuscript.
>
> ## Reporting PPL as a metric
>
> We agree with the reviewer that calling our reported result perplexity on DNA is inaccurate. To compute our metric for DNA sequences we used the continuous time change of variable of the flow along the ODE trajectory over the manifold to compute the log-likelihood of the observed sequence (as done in Riemannian Flow Matching). Since the output space is discrete this likelihood on the manifold is an upper bound to the discrete log likelihood. More accurately this is a valid Evidence Lower Bound (ELBO) which was proven in concurrent masked discrete diffusion models [1, 2]. We also note that for autoregressive models the used PPL metric is a specific factorisation over the joint $\log p(x_{1:N})$ and any valid ELBO is comparable. Nevertheless, the reviewer is correct and we will amend our reported results by indicating this upperbound when reporting PPL for diffusion/flow matching models.
>
> Finally, we have also included a new Gen PPL eval metric for DNA sequences. This is the PPL supplied a larger pre-trained autoregressive model on the outputs samples of a generative model. We found this metric could not distinguish the performance of DFM and Fisher-Flows, but we report it for completeness.
> ## Q1
> Please see our global response for new larger scale experiments, including on text.
> ## Q2
> Please see above about additional baselines as well as our global response.
> ## Q3
> Please see our response above which finds flaws in the empirical setup of DFM.
> ## Q4
> We computed an upper bound to the PPL using the continuous time change of variable formula associated with an ODE. This is a valid ELBO see [1-2] for proof.
> ## Q5
> OT in Fisher-Flows is not computationally expensive as both the sphere and the simplex have closed form distance functions which allows us to efficiently calculate the OT plan over mini-batches. This adds minimal overhead to training and is not affected by scaling.
> ## Q6
> Thank you for pointing this out. We will fix it in the updated paper.
> ## Conclusion
> We thank the reviewer again for their time. We believe we have answered to the best of our ability all the great questions raised by the reviewer. We hope our answers allow the reviewer to consider potentially upgrading their score if they see fit. We are also more than happy to answer any further questions.
>
> ## References
> [1] Sahoo et al. "Simple and Effective Masked Diffusion Language Models." arXiv preprint arXiv:2406.07524 (2024).
>
> [2] Shi et al. "Simplified and Generalized Masked Diffusion for Discrete Data." arXiv preprint arXiv:2406.04329 (2024).

---

> > ### Comment · Reviewer_11qA · 2024-08-11
> >
> > I thank the authors for their efforts in addressing my concerns and questions. Specifically, I appreciate that the authors have followed my suggestions to add additional experiments in other domains including graph generation and text generation. The additional results look interesting. Specifically, the LM1B results look somewhat promising, although it seems the results on graph generation are not very strong. However, my concerns regarding the validity of the DNA design benchmark still remain.
> >
> > ## Regarding PPL calculation
> >
> > I appreciate that the authors have provided an alternative PPL derived from recent concurrent work. I noticed the MDLM paper relies on masked diffusion, so I would appreciate further clarifications from the authors on the calculation of PPL. Specifically, did you calculate the NLL according to Equation 10 in the MDLM paper as
> > $$
> > \mathcal{L}\_\text{NELBO}=\mathbb{E}\_q\int\_{t=0}^{t=1}-\frac{1}{1-t}\log\langle x\_\theta(z\_t,t),x\rangle dt
> > $$
> > How did you choose the noised data $z_t$? Did you use the standard Euclidean inner product or the Fisher inner product defined in your Eq.5?
> >
> > ## Regarding Baselines for DNA Design
> >
> > The authors' response to my concerns regarding the baseline results for DNA design does not seem to have solid grounds, for which I further elaborate as follows.
> >
> > >  the provided command used --subset_train_as_val argument which sets the validation set as the training set.  This meant that their reported metrics were measured on the training set.
> >
> > This is true for enhancer design, but not true for promoter design in which the model is evaluated on the test set (according to the command in the DFM repo).
> >
> >
> > > Their synthetic data experiment did not standardise a training and validation split and meant that their training set size was a function of run time—effectively infinite. Operationally, DFM was trained for $10^{10}$ points per epoch which is a tremendous amount for a synthetic task. We standardised the experiment to train all models with a more modest 10^6 points.
> >
> > This is the setting for the toy experiments but not for promoter design & enhancer design. According to the DFM paper Appendix B.1, it was trained for 200 epochs for promoter design and 800 epochs for enhancer design. In FisherFlow, Appendix F.4 indicated it was trained for $200000 \times 256 / 88470 \approx 579$ epochs for promoter design and $450000 \times 256 / 70892 \approx 1625$ epochs for enhancer design. In both cases, FisherFlow was trained on a doubling of data.
> >
> > Moreover, **the authors didn't respond to my concerns regarding the discrepancy of the results in the promoter design task**. To give a more concrete evaluation, I tried the DFM repo to test their provided checkpoint on the test set. **Five repeated experiments give an MSE of 0.02697 ± 0.00024**, which matches the reported number in the original DFM paper but is significantly smaller than the reported number in this paper. Furthermore, the new results for the Linear FM baseline in the authors' rebuttal are significantly worse than those reported in the DFM paper (despite that, the Linear FM still achieves better PPL than FisherFlow). Given that the DFM checkpoint matches the reported MSE, I would consider the result in the DFM paper for Linear FM to be more credible.
> >
> > It is also unclear, as I have mentioned in my review, why the authors reran the experiments *only* for DFM but copied the results for all the other baseline models directly from DFM for promoter design. If the authors decided to use an alternative experimental setting, they should rerun all the baselines to be consistent and comparable.
> >
> > In conclusion, I think the authors did not provide a convincing explanation regarding the discrepancy in the results of the promoter design task. The reported results for this task do not seem credible to me.

---

> > > ### Author Response · Authors · 2024-08-11
> > >
> > > We thank the reviewer for engaging with us in this rebuttal and for their recommendations that improve the empirical caliber of this paper. We now answer the questions raised in their response.
> > >
> > > ## PPL Calculation
> > >
> > > For our new experiments, note that we outlined how the intermediate noisy distribution, $p_t$, is constructed, in the global response. This $z_t = x_t \sim p_t$ is a convex combination of 2 paths, a noisy path from a masked prior to $x_0$, and a noisy path from a uniform prior also to $x_0$. Each path for us uses the Fisher geodesic. Note that for a masked state to $x_0$ we traverse the edge of the sphere along its natural geodesic automatically.
> > >
> > > >Did you use the standard Euclidean inner product or the Fisher inner product defined in your Eq.5?
> > >
> > > To compute the NLL, we use the provided MDLM code which computes the NELBO as $-\log p(x_0 | x_t)$ integrated across time. We use the Fisher inner-product on the Sphere in eqn 10 of MDLM to compute $<x_{\theta}(x_t, t) , x_0>$.
> > >
> > > ## Further clarifications on DNA experiments
> > >
> > > We appreciate the reviewer's efforts in helping increase the empirical transparency of our DNA experiments and that our code is also included in this submission. Moreover, **we commit to releasing all of our DNA experiment code (already included in the supplementary zip file) to enhance reproducibility.**
> > >
> > > We thank the reviewer for agreeing on the experimental issues in the DFM code for the synthetic and DNA Enhancer experiments. The former was trained on an unreasonable amount of data; the latter was trained and evaluated on the training set due to the use of `--subset_train_as_val` flag. These issues in different experiments we argue justify our decision to build a more rigorous setup for all experiments, including DNA Promoter.
> > >
> > > >I tried the DFM repo to test their provided checkpoint on the test set. Five repeated experiments give an MSE of 0.02697 ± 0.00024,
> > >
> > > We appreciate the reviewer taking the time to rerun DFM experiments in their original codebase with their checkpoint. Our reproduction inherits the exact same model architecture and code for DFM and our experimental protocol primarily addresses the dataset and evaluation pipelines. Their public code only provides **a single trained checkpoint** and, as a result, their reported MSE in Tab 1. of 0.0269 does not contain stds over runs. We assume that the reviewer reran the Promoter experiments with 5 seeds using this checkpoint which quantifies the performance of DFM only on this trained checkpoint, *with the randomness being due to the sampling of the prior*. In contrast, our DFM reproduction retrains the model from scratch 5 times and then evaluates this method. We argue that this is a more robust experimental setup as the randomness is over the entire training procedure and limits the cherry-picking of a preferred run/checkpoint. Our top seed matched the original reported result in DFM but the mean over 5 seeds is worse. Consequently, we attribute the difference in reported numbers due to randomness over 5 runs vs. their 1 run.
> > >
> > > We hope our answer here provides more clarity and credibility to the reproduction of our DNA Promoter experiments.
> > >
> > > >[...] FisherFlow was trained on a doubling of data.
> > >
> > > Since we harmonized DFM, Linear, and Fisher-Flow implementations in our codebases both DFM and Fisher Flows are trained on the exact same amount of data. We evaluated all models on the test set using the best validation MSE (Appendix F.4.2), so the total quantity of data seen as a function of epochs is, very respectfully, not a valid concern.
> > >
> > > >  Linear FM still achieves better PPL than FisherFlow
> > >
> > > We note that the std of Fisher-Flows overlaps with LinearFM in our 1pg PDF as such it is within the margin of error. Given the limited time frame of the rebuttal, we only had time to run 1 training seed for LinearFM but we will include 5 seeds in our updated paper.
> > >
> > > > Why the authors reran the experiments only for DFM but copied the results for all the other baseline models directly from DFM
> > >
> > > We found in the original DFM paper (Tab. 2) that the promoter baseline numbers, outside of the language model, were taken from [1] and, as a result, felt that these baselines were decoupled enough from the DFM codebase that they did not suffer the same flaws in the experimental setup.
> > > We reran the LM, Linear, and DFM baselines in our experimental setup as they were included in their original DFM codebase which gave us an opportunity to report 5 seeds. We also note the other baselines were not included in the original DFM codebase. We will update our paper to include the origin of all baselines.
> > >
> > > We hope our response here alleviates the concerns raised by the reviewer and allows the reviewer to endorse our paper more strongly. We are also happy to answer any further questions, please let us know.
> > >
> > > [1] Avdeyev, Pavel, et al. "Dirichlet diffusion score model for biological sequence generation." International Conference on Machine Learning. PMLR, 2023.

---

> > > ### Author Response · Authors · 2024-08-12
> > > **Kindly awaiting further discussion**
> > >
> > > Dear Reviewer,
> > >
> > > We thank you for all of your time and effort in engaging with us during this rebuttal period. As the deadline for this period is fast approaching we were wondering if there were any further clarifications needed from our responses so that we share the same view regarding the chief concern regarding the reproduction of the DNA promoter experiments and or the new experiments added in the global response. We highlight the following facts that may aid clarity and transparency: 1.) Our DFM reproduction and Fisher-Flows were trained in the same setup and we used validation MSE to select the best model to evaluate. So no models saw *more data* 2.) Our experiments used 5 random seeds of the model's retrained from scratch each time and our best DFM result matched the reported paper and released checkpoint result of DFM, but the mean was higher 3.) All other baselines were taken from Ayedev et. al 2023, which is also the case for DFM.
> > >
> > > We hope our rebuttal responses have helped strengthen the reviewer's view on our experiments and we are more than happy to engage in any further discussion before the end of the rebuttal period. We thank the reviewer again for their time and if our responses and new experiments merit it we would also be appreciative if the reviewer would kindly consider a fresh evaluation of our work.

---

> > > > ### Comment · Reviewer_11qA · 2024-08-13
> > > > **Decision to raise my score**
> > > >
> > > > I first thank the authors for their clarifications on the PPL calculation following MDLM. Regarding my previous concerns about the promoter design task, I took the time to reran one promoter design experiment of DFM using the code they provided (apologies for the delayed response). I obtained a test MSE of 0.02911, which was larger than the provided checkpoint in the DFM repo but still smaller than that claimed in this paper. Nonetheless, given that the authors mentioned the reported MSE for DFM came from 5 repeated experiments (instead of 5 repeated evaluations on the generated sequences) and considering the possible randomness based on my reproduction of DFM, although my reproduction of DFM still indicated a better performance, I reckon that the promoter design task may not be the best task to demonstrate the effectiveness of different models. Still, I highly suggest the authors revisit their experiment, open source code that was used for benchmarking DFM, and explain the discrepancy of DFM's result clearly to improve the credibility of the result.
> > > >
> > > > Based on the above observations, although I still hold that DFM performs better in the promoter design task, I decided to put less emphasis on this task. The authors primarily focused on DNA design tasks in the original manuscript and extended the proposed model to additional tasks in their rebuttal. As I have noted in my previous comment, the LM1B results look more promising whereas the results on graph generation are not very strong. Despite this, I do acknowledge the novelty of applying the Fisher-Rao metric and believe mathematically this work can provide an interesting aspect to the generative community. In this regard, I raise my score from 4 to 5.

---

> > > > > ### Author Response · Authors · 2024-08-13
> > > > > **Re: Decision to raise my score**
> > > > >
> > > > > We thank the reviewer for taking additional time to investigate DFM on DNA Promoter. We will most definitely open-source our code and include a detailed discussion regarding the discrepancy between our DFM reproduction and the published table results.
> > > > >
> > > > > We appreciate that the reviewer views our new experimental findings on LM1B as promising and that the reviewer finds using that applying the Fisher-Rao metric to be novel and interesting to the generative community. Finally, we thank the reviewer again for their effort in helping us strengthen our paper and we will incorporate all the great suggestions made by the reviewer in our updated draft.

---

### Author Rebuttal · Authors · 2024-08-07

We thank all the reviewers for their time and constructive comments. We are grateful that the reviewers appreciated the novelty of our geometric perspective to discrete generative modelling (R 11qA) and that it is a natural application of Riemannian flow matching over simplices (R cNL3). We are also heartened to hear that the reviewers agree that our parametrization of flows over the sphere is beneficial for numerical stability and leading to demonstrated empirical benefits with little computation overhead (R 11qA, YHTA).  We now address shared concerns raised by the reviewers, and summarise our new experiments and ablations included in the supplementary PDF.
## New experiments on complex data (R 11qA, R cNL3)
The reviewers expressed concerns regarding the empirical verification of Fisher-Flows against other domains. We included, in our rebuttal PDF, 2 additional standard discrete generative modelling tasks: molecule generation (QM9), and language modelling on the One Billion Words Dataset (LM1B). These new domains enable us to test the scalability of Fisher-Flows against SOTA discrete generative models for molecules and text.
### Experimental setup and Flow parameterization
To enable scalable and effective training of our flows, we parametrize the loss using the target prediction framework by learning a denoising network to predict the target discrete target token given $x_t$. We note that the vector field can easily be recovered for inference and used to predict the next Euler step—thus no extra cost is added to sampling.
### Language Modelling
We train Fisher-Flows on the LM1B dataset, which has a vocabulary size of ~800k words. This experiment seeks to address the scalability concerns as well as enable direct comparison to a more standard discrete generative modelling setting.

We define Fisher-Flows by using the following probability path $p_t = \kappa_t * p_{M} + (1 - kappa_t) * p_{unif}$, where $\kappa_t \in [0,1]$. Here $p_{M}$ is the Fisher probability path between the target $x_0$ and designated mask token $M$, while $p_{unif}$ is also a Fisher probability path from a sample from a uniform distribution and $x_0$. Thus $p_t$ is a convex combination of probability paths using a noise scheduler $\kappa_t$. Using a denoising architecture enables us to rewrite the original loss as a weighted negative log likelihood $L = \mathbb{E}[-\log p(x_0 | x_t)]]$. Crucially, this allows us to calculate an upper bound to the test perplexity, which is a natural evaluation metric for language modelling and employed by concurrent SOTA works on discrete diffusion MDLM and MD4 [1, 2], who prove that this is a valid Evidence Lower Bound.

We report our results in Table 2 of the 1pg PDF and note that Fisher-Flows is the best performing discrete diffusion/flow-matching model achieving a test perplexity of $\leq 26.51$ and $\leq 22.42$ after training on $33B$ and $327B$ tokens respectively. We note that our method outperforms the baselines suggested by the reviewer of D3PM, SEDD, as well as marginally outperforming concurrent masked diffusion MDLM [1].
### Molecule Generation
We instantiate Fisher-Flows on QM9 where four different data types must be produced: positions, charges, atom types, and the possible bonds between them. Furthermore, the positions are continuous rather than discrete which allows us to showcase the feasibility of using Fisher-Flows in a mixed setting of discrete and continuous data.

We report our results in Table 3 of the rebuttal PDF. We find that Fisher Flows outperforms a comparable flow matching baseline in EquiFM [3] and is competitive with a diffusion baseline in Jodo [4], all saturating the benchmark. We also include examples of generated samples in Fig. 2 from Fisher Flows to qualitatively evaluate our approach.
## New ablations and evaluation metrics (R 11qA, RcNL3, YHTA)
To make our comparisons more robust, we extended the synthetic experiment with a variety of new baselines; the results are shown in Fig 1 of the rebuttal PDF. We included Linear Flow matching, D3PM, and Multinomial Flow, as suggested by the reviewers. Fisher-Flows clearly outperform Linear FM across all dimensions. Although between dimensions 40 and 100/120 Fisher-Flow performs marginally worse than both D3PM and Multinomial Flows, past dimensions 100 and 120 these two’s KL divergences diverge to infinity (hence their absence from the figure) indicating that some categories have been sampled essentially zero times. Our Fisher-Flow parametrization does not incur this numerical failure mode.
### Generative Perplexity
We updated our DNA sequence results in Table 1 of the rebuttal PDF by first amending the reported Perplexity as an upper bound to the true test perplexity, as our flows yield a valid ELBO. In addition, we ålso included a common evaluation method (Gen-PPL) in the perplexity induced by autoregressive language models that we train on the same dataset as the evaluated generative model. We train small GPT2 DNA language models on respective datasets and assess the flow model’s DNA generations under these. Random sequences obtain gen-ppls of over 4 for Promoter DNA, fly-brain and melanoma DNA LMs. The gen-ppl values for Fisher-Flows, Dirichlet Flow Matching and Linear Flow matching for all DNA datasets are all around 1, exhibiting no clear winner under this metric.

We also include a Linear Flow Matching baseline and find that Fisher Flows is on par and both are better than Dirichlet Flow Matching.
## References
[1] Sahoo et al. "Simple and Effective Masked Diffusion Language Models." arXiv preprint arXiv:2406.07524 (2024).

[2] Shi et al. "Simplified and Generalized Masked Diffusion for Discrete Data." arXiv preprint arXiv:2406.04329 (2024).

[3] Song  et al. "Equivariant flow matching with hybrid probability transport." arXiv preprint arXiv:2312.07168 (2023).

[4] Huang et al. "Learning joint 2d & 3d diffusion models for complete molecule generation." arXiv preprint arXiv:2305.12347 (2023).

---

### Decision · Program_Chairs · 2024-09-25

**Decision:**

Accept (poster)

**Comment:**

This paper proposes a flow matching method for generative modeling of discrete data. The authors proceed by operating on the (continuous) simplex of discrete probability distributions, where they use Riemannian flow matching along with the Fisher-Rao metric. The authors also propose to map the simplex onto the positive orthant of a hypersphere, which they argue improves performance.

The authors provided a thorough rebuttal addressing concerns by the reviewers, who now all unanimously agree that this paper meets the bar for publication at NeurIPS. I also agree and thus recommend acceptance. For the camera-ready version of the paper, I encourage the authors to include the additional experiments and ablations carried out during the rebuttal, and to more carefully articulate the empirical benefits of using the sphere map.